# Improved Active Learning via Dependent Leverage Score Sampling

**Atsushi Shimizu, Xiaoou Cheng, Christopher Musco, Jonathan Weare**
New York University
{as15106,chengxo,cmusco,weare}@nyu.edu

## Abstract

We show how to obtain improved active learning methods in the agnostic (adversarial noise) setting by combining marginal leverage score sampling with non-independent sampling strategies that promote spatial coverage. In particular, we propose an easily implemented method based on the *pivotal sampling algorithm*, which we test on problems motivated by learning-based methods for parametric PDEs and uncertainty quantification. In comparison to independent sampling, our method reduces the number of samples needed to reach a given target accuracy by up to $50\%$. We support our findings with two theoretical results. First, we show that any non-independent leverage score sampling method that obeys a weak *one-sided $\ell_\infty$ independence condition* (which includes pivotal sampling) can actively learn $d$ dimensional linear functions with $O(d \log d)$ samples, matching independent sampling. This result extends recent work on matrix Chernoff bounds under $\ell_\infty$ independence, and may be of interest for analyzing other sampling strategies beyond pivotal sampling. Second, we show that, for the important case of polynomial regression, our pivotal method obtains an improved bound on $O(d)$ samples.

## 1 Introduction

In the active linear regression problem, we are given a data matrix $\mathbf{A} \in \mathbb{R}^{n \times d}$ with $n \gg d$ rows and query access to a target vector $\mathbf{b} \in \mathbb{R}^n$. The goal is to learn parameters $\mathbf{x} \in \mathbb{R}^d$ such that $\mathbf{A}\mathbf{x} \approx \mathbf{b}$ while observing as few entries in $\mathbf{b}$ as possible. We study this problem in the challenging agnostic learning or "adversarial noise" setting, where we do not assume any underlying relationship between $\mathbf{A}$ and $\mathbf{b}$. Instead, our goal is to find parameters competitive with the best possible fit, good or bad. Specifically, considering $\ell_2$ loss, let $\mathbf{x}^* = \arg\min_{\mathbf{x}} \|\mathbf{A}\mathbf{x} - \mathbf{b}\|_2^2$ be optimal model parameters. We want to find $\tilde{\mathbf{x}}^*$ using a small number of queried target values in $\mathbf{b}$ such that

$$\|\mathbf{A}\tilde{\mathbf{x}}^* - \mathbf{b}\|_2^2 \le (1 + \epsilon)\|\mathbf{A}\mathbf{x}^* - \mathbf{b}\|_2^2, \tag{1.1}$$

for some error parameter $\epsilon > 0$. Beyond being a fundamental learning problem, active regression has emerged as a fundamental tool in learning based methods for the solution and uncertainty analysis of parametric partial differential equations (PDEs) (Chkifa et al., 2015; Guo et al., 2020). For such applications, the agnostic setting is crucial, as a potentially complex quantity of interest is approximated by a simple surrogate model (e.g. polynomials, sparse polynomials, single layer neural networks, etc.) (Lüthen et al., 2021; Hokanson and Constantine, 2018). Additionally, reducing the number of labels used for learning is crucial, as each label usually requires the computationally expensive numerical solution of a PDE for a new set of parameters (Cohen and DeVore, 2015).

### 1.1 Leverage Score Sampling

Active linear regression has been studied for decades in the statistical model where $\mathbf{b}$ is assumed to equal $\mathbf{A}\mathbf{x}^*$ plus i.i.d. random noise. In this case, the problem can be addressed using tools from optimal experimental design (Pukelsheim, 2006). In the agnostic case, near-optimal sample complexity results were only obtained relatively recently using tools from non-asymptotic matrix concentration (Tropp, 2012). In particular, it was shown independently in several papers that collecting entries from $\mathbf{b}$ *randomly* with probability proportional to the *statistical leverage scores* of rows in $\mathbf{A}$ can achieve (1.1) with $O(d \log d + d/\epsilon)$ samples (Sarlós, 2006; Rauhut and Ward, 2012; Hampton and Doostan, 2015; Cohen and Migliorati, 2017). The leverage scores are defined as follows:

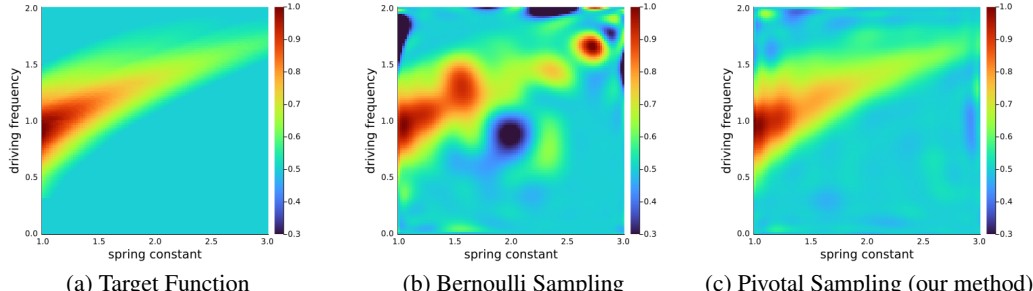

| (a) Target Function | (b) Bernoulli Sampling | (c) Pivotal Sampling (our method) |

Figure 1: Polynomial approximations to the maximum displacement of a damped harmonic oscillator, as a function of driving frequency and spring constant. (a) is the target value, and samples can be obtained through the numerical solution of a differential equation governing the oscillator. Both (b) and (c) draw 250 samples using leverage score sampling and perform polynomial regression of degree 20. (b) uses Bernoulli sampling while (c) uses our pivotal sampling method. Our method gives a better approximation, avoiding artifacts that result from gaps between the Bernoulli samples.

**Definition 1.1** (Leverage Score). *Let $\mathbf{U} \in \mathbb{R}^{n \times r}$ be any orthogonal basis for the column span of a matrix $\mathbf{A} \in \mathbb{R}^{n \times d}$. Let $\mathbf{a}_i$ and $\mathbf{u}_i$ be the $i$-th rows of $\mathbf{A}$ and $\mathbf{U}$, respectively. The leverage score $\tau_i$ of the $i$-th row in $\mathbf{A}$ can be equivalently written as:*

$$\tau_i = \|\mathbf{u}_i\|_2^2 = \mathbf{a}_i^T (\mathbf{A}^T \mathbf{A})^{-1} \mathbf{a}_i = \max_{\mathbf{x} \in \mathbb{R}^d} (\mathbf{a}_i^T \mathbf{x})^2 / \|\mathbf{A}\mathbf{x}\|_2^2. \tag{1.2}$$

*Notice that $\tau_i = \|\mathbf{u}_i\|_2^2$, $\sum_{i=1}^n \tau_i = d$ when $\mathbf{A}$ is full-rank and thus $\mathbf{U}$ has $r = d$ columns.*

See (Avron et al., 2017) for a short proof of the final equality in (1.2). This last definition, based on a maximization problem, gives an intuitive understanding of the leverage scores. The score of row $\mathbf{a}_i$ is higher if it is more "exceptional", meaning that we can find a vector $\mathbf{x}$ that has large inner product with $\mathbf{a}_i$ relative to its average inner product (captured by $\|\mathbf{A}\mathbf{x}\|_2^2$) with all other rows in the matrix. Based on leverage score, rows that are more exceptional are sampled with higher probability.

Prior work considers independent leverage score sampling, either with or without replacement. The typical approach for sampling without replacement, which we call "Bernoulli sampling" is as follows: Each row $\mathbf{a}_i$ is assigned a probability $p_i = \min(1, c \cdot \tau_i)$ for an oversampling parameter $c \geq 1$. Then each row is sampled independently with probability $p_i$. We construct a subsampled data matrix $\tilde{\mathbf{A}}$ and subsampled target vector $\tilde{\mathbf{b}}$ by adding $\mathbf{a}_i/\sqrt{p_i}$ to $\tilde{\mathbf{A}}$ and $b_i/\sqrt{p_i}$ to $\tilde{\mathbf{b}}$ for any index $i$ that is sampled. To solve the active regression problem, we return $\tilde{\mathbf{x}}^* = \arg\min_{\mathbf{x}} \|\tilde{\mathbf{A}}\mathbf{x} - \tilde{\mathbf{b}}\|_2$.

### 1.2 OUR CONTRIBUTIONS

In applications to PDEs, the goal is often to approximate a function over a low dimensional distribution $\mathcal{X}$. E.g. $\mathcal{X}$ might be uniform over an interval $[-1, 1] \subset \mathbb{R}$ or over a box $[-1, 1] \times \ldots \times [-1, 1] \subset \mathbb{R}^q$. In this setting, the length $d$ rows of $\mathbf{A}$ correspond to feature transformations of samples from $\mathcal{X}$. For example, in the ubiquitous task of polynomial regression, we start with $\mathbf{x} \sim \mathcal{X}$ and add to $\mathbf{A}$ a row containing all combinations of entries in $\mathbf{x}$ with total degree $p$, i.e., $x_1^{\ell_1} x_2^{\ell_2} \ldots x_q^{\ell_q}$ for all non-negative integers $\ell_1, \ldots, \ell_q$ such that $\sum_{i=1}^q \ell_i \leq p$. For such problems, "grid" based interpolation is often used in place of randomized methods like leverage scores sampling, i.e., the target $\mathbf{b}$ is queried on a deterministic grid tailored to $\mathcal{X}$. For example, when $\mathcal{X}$ is uniform on a box, the standard approach is to use a grid based on the Chebyshev nodes (Xiu, 2016). Pictured in Figure 2, the Cheybshev grid concentrates samples near the boundaries of the box, avoiding the well known issue of Runge's phenomenon for uniform grids. Leverage score sampling does the same. In fact, the methods are closely related: in the high degree limit, the leverage scores for polynomial regression over the box match the asymptotic density of the Chebyshev nodes (Lüthen et al., 2021).

So how do the deterministic and randomized methods compare? The advantage of randomized methods based on leverage score sampling is that they yield strong provable approximation guarantees, and easily generalize to any distribution $\mathcal{X}$.[1] Deterministic methods are less flexible on the

---

[1]As discussed in (Chen and Price, 2019), no deterministic method can provably solve the agnostic regression problem with few samples. Since we make no assumptions on $\mathbf{b}$, all error in $\mathbf{A}\mathbf{x}^* - \mathbf{b}$ could be concentrated only at the deterministic indices to be selected. Randomization is needed to avoid high error outliers in $\mathbf{b}$.

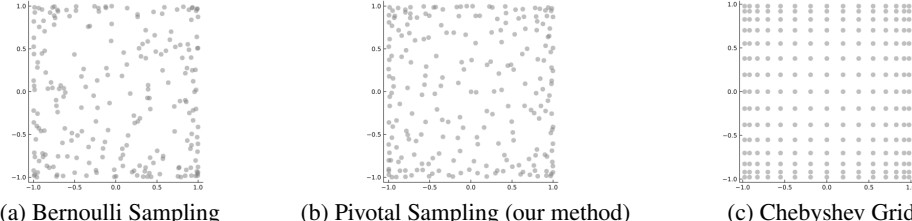

|(a) Bernoulli Sampling | (b) Pivotal Sampling (our method) | (c) Chebyshev Grid|

Figure 2: The results of three different active learning methods used to collect samples to fit a polynomial over $[-1, 1] \times [-1, 1]$. The image on the left was obtained by collecting points independently at random with probability according to their statistical leverage scores. The image on the right was obtained by collecting samples at the 2-dimensional Chebyshev nodes. The image in the middle shows our method, which collects samples according to leverage scores, but using a non-independent pivotal sampling strategy that ensures samples are evenly spread in spatially.

other hand, and do not yield provable guarantees. However, the advantage of grid based methods is that they more "evenly" distribute samples over the original data domain, which can lead to better performance in practice. Randomized methods are prone to "missing" larger regions of $\mathcal{X}$'s support, as shown in Figure 2. The driving question behind our work is:

> *Is it possible to obtain the "best of both worlds" for fitting functions over low-dimensional domains? I.e., can we match or improve on the strong theoretical guarantees of leverage score sampling with a method that produces spatially well-distributed samples?*

We answer this question in the affirmative. Instead of sampling rows from $\mathbf{A}$ *independently* with probability proportional to the leverage scores, we adopt a tool from survey sampling known as pivotal sampling (Deville and Tille, 1998). Our specific version of pivotal sampling is *spatially-aware*, meaning that it covers the domain in a well-balanced manner, while the marginal probabilities remain proportional to the leverage score. At a high-level, the pivotal method is a "competition" based sampling approach, where candidate rows compete in a binary tree tournament. By structuring the tournament so that spatially close points compete at lower levels (we use a novel recursive PCA procedure to build the tree), we ensure better spatial spread than Bernoulli leverage score sampling.

We show that our pivotal sampling method matches or beats the complexity of independent leverage score sampling in theory, while performing significantly better in practice. On the practice side, we offer Figure 1 as an example from a PDE test problem. In comparison to independent sampling, our spatially-aware method obtains a much better approximation to the target for a fixed number of samples (more details in Section 4). On the theory side, we prove two results. The first is general: we show that, as long as it samples rows from $\mathbf{A}$ with *marginal probabilities* proportional to the leverage scores, *any* sampling strategy that obeys a weak "one-sided $\ell_\infty$ independence" condition (which includes pivotal sampling) matches the complexity of independent leverage score sampling:

**Theorem 1.1.** *Let $\mathbf{A} \in \mathbb{R}^{n \times d}$ be a data matrix and $\mathbf{b} \in \mathbb{R}^n$ be a target vector. Consider any algorithm which samples exactly $k$ rows from $\mathbf{A}$ (and observes the corresponding entries in $\mathbf{b}$) from a distribution that 1) satisfies one-sided $\ell_\infty$ independence (Defn. 3.1) with parameter $D_{inf}$ and 2) the marginal probability of sampling any row $\mathbf{a}_i$ is proportional to $\tau_i$. [2] Let $\tilde{\mathbf{A}}$ and $\tilde{\mathbf{b}}$ be the scaled sampled data and target, as defined in Section 1.1, and let $\tilde{\mathbf{x}}^* = \arg\min_{\mathbf{x} \in \mathbb{R}^d} \|\tilde{\mathbf{A}}\mathbf{x} - \tilde{\mathbf{b}}\|_2^2$. As long as $k \geq c \cdot \left( d \log d \cdot D_{inf}^2 + \frac{d}{\epsilon} \cdot D_{inf} \right)$ for a fixed positive constant $c$, then with probability $99/100$,*

$$\|\mathbf{A}\tilde{\mathbf{x}}^* - \mathbf{b}\|_2^2 \leq (1 + \epsilon)\|\mathbf{A}\mathbf{x}^* - \mathbf{b}\|_2^2 \tag{1.3}$$

One-sided $\ell_\infty$ independence was introduced in a recent paper by Kaufman et al. (2022) on matrix Chernoff bounds (we provide a formal definition in Sec. 3). It is the *weakest* condition under which a tight matrix Chernoff bound is known to hold. For example, the condition is implied with constant $D_{inf} = O(1)$ by almost all existing notions of negative dependence between random variables, including conditional negative association (CNA) and the strongly Rayleigh property (Pemantle, 2000). As discussed in Sec. 3, a tight matrix Chernoff bound is a prerequisite for proving relative

---

[2]Formally, we assume that the marginal probability of sampling $\mathbf{a}_i$ equals $\min(1, c\tau_i)$ for a fixed constant $c \geq 1$. Our proof easily generalizes to the case when some probabilities exceed this bound (since sampling more never hurts) although the total sample complexity will depend on the sum of the marginal probabilities.

error active learning results like Equation (1.3), however does not imply such a result alone. Our proof of Theorem 1.1 requires adapting an approximate matrix-multiplication method from (Drineas et al., 2006) to non-independent sampling. It can be viewed as extending the work of Kaufman et al. (2022) to show that essentially all sampling distributions known to yield tight matrix Chernoff bounds also yield near optimal active regression bounds in the agnostic setting. Importantly, this includes binary-tree-based pivotal sampling methods like those introduced in this work. Such methods are known to satisfy the strongly Rayleigh property (Brändén and Jonasson, 2012), and thus one-sided $\ell_\infty$ independence with $D_{\inf} = O(1)$. So, as a corollary of Theorem 1.1, we obtain:

**Corollary 1.1.** *The spatially-aware pivotal sampling methods introduced in Section 2 (which use a fixed binary tree) return with probability $99/100$ a vector $\tilde{\mathbf{x}}^*$ satisfying $\|\mathbf{A}\tilde{\mathbf{x}}^* - \mathbf{b}\|_2^2 \leq (1 + \epsilon)\|\mathbf{A}\mathbf{x}^* - \mathbf{b}\|_2^2$ while only observing $O\left(d \log d + \frac{d}{\epsilon}\right)$ entries in $\mathbf{b}$.*

We hope that Theorem 1.1 will be valuable in obtaining similar results for other sampling methods beyond our own. However, the result falls short of justifying why pivotal sampling performs *better* than independent leverage score sampling in experiments. Towards that end, we prove a second result specific to pivotal sampling, which shows that the method actually improves on the complexity of independent sampling by a log factor in the important special case of polynomial regression:

**Theorem 1.2.** *Consider any function $b : [\ell, u] \to \mathbb{R}$ defined on an interval $[\ell, u] \subset \mathbb{R}$, and consider fitting $b$ with a degree $d$ polynomial based on evaluations of the function at $x_1, \ldots, x_k \in [\ell, u]$. If $x_1, \ldots, x_k$ are collected via pivotal sampling with leverage score marginals (see Appendix C for details), then as long as $k \geq c \cdot (d + \frac{d}{\epsilon})$ for a fixed positive constant $c$, there is a procedure that uses these samples to construct a degree $d$ polynomial $\tilde{p}$ which, with probability $99/100$, satisfies:*

$$\|\tilde{p} - b\|_2^2 \leq (1 + \epsilon) \min_{\text{degree } d \text{ polynomial } p} \|p - b\|_2^2.$$

*Here $\|f\|_2^2$ denotes the average squared magnitude $\int_\ell^u f(x)^2 dx$ of a function $f$.*

The problem of finding a polynomial approximation to a real-valued function that minimizes the average square error $\|p - b\|_2^2$ can be modeled as an active regression problem involving a matrix $\mathbf{A}$ with $d + 1$ columns and an infinite number of rows. In fact, polynomial approximation is one of the primary applications of prior work on leverage score-based active learning methods (Cohen and Migliorati, 2017; Avron et al., 2019). Such methods require $O\left(d \log d + \frac{d}{\epsilon}\right)$ samples, so Theorem 1.2 is better by a $\log d$ factor. Since polynomial approximation is a representative problem where spatially-distributed samples are important, Theorem 1.2 provides theoretical justification for the strong performance of pivotal sampling in experiments. Our proof is inspired by a result of Kane et al. (2017), and relies on showing a tight relation between the leverage scores of the polynomial regression problem and the orthogonality measure of the Chebyshev polynomials on $[\ell, u]$.

### 1.3 RELATED WORK

The application of leverage score sampling to the agnostic active regression problem has received significant recent attention. Beyond the results discussed above, extensions of leverage score sampling have been studied for norms beyond $\ell_2$ (Chen and Derezinski, 2021; Musco et al.; Meyer et al., 2023; Parulekar et al., 2021), in the context where the sample space is infinite (i.e. $\mathbf{A}$ is an operator with infinite rows) (Erdélyi et al., 2020; Avron et al., 2019), and for functions that involve non-linear transformations (Gajjar et al., 2023; Mai et al., 2021; Munteanu et al., 2018).

Theoretical improvements on leverage score sampling have also been studied. Notable is a recent result that improves on the $O(d \log d + d/\epsilon)$ bound by a $\log d$ factor, showing that the active least squares regression problem can be solved with $O(d/\epsilon)$ samples (Chen and Price, 2019). This is provably optimal. However, the algorithm in (Chen and Price, 2019) is complex, and appears to involve large constant factors: in our initial experiments, it did not empirically improve on independent leverage score sampling. In contrast, by Theorem 1.2, our pivotal sampling method matches the theoretical sample complexity of (Chen and Price, 2019) for the special case of polynomial regression, but performs well in experiments (significantly better than independent leverage score sampling). There have been a few other efforts to develop practical improvements on leverage score sampling. Similar to our work, Dereziński et al. (2018) study a variant of volume sampling that matches the theoretical guarantees of leverage score sampling, but performs better experimentally. However, this method does not explicitly take into account spatial-structure in the underlying regression problem. While the method from (Dereziński et al., 2018) does not quite fit our Theorem 1.1

(e.g., it samples indices *with* replacement) we expect similar methods could be analyzed as a special case of our result, as volume sampling induces a strongly Rayleigh distribution.

While pivotal sampling has not been studied in the context of agnostic active regression, it is widely used in other applications, and its negative dependence properties have been studied extensively. (Dubhashi et al., 2007) proves that pivotal sampling satisfies the negative association (NA) property. (Borcea et al., 2009) introduced the notion of a strongly Rayleigh distribution and proved that it implies a stronger notion of conditional negative association (CNA), and (Brändén and Jonasson, 2012) showed that pivotal sampling run with an arbitrary binary tree is strongly Rayleigh. It follows that the method satisfies CNA. (Greene et al., 2022) discusses an efficient algorithm for pivotal sampling by parallelization and careful manipulation of inclusion probabilities. Another variant of pivotal sampling that is *spatially-aware* is proposed in (Grafström et al., 2011). Though their approach is out of the scope of our analysis as it involves randomness in the competition order used during sampling, our method is inspired by their work.

### 1.4 NOTATION AND PRELIMINARIES

**Notation.** We let $[n]$ denote $\{1, \cdots, n\}$. $\mathbb{E}[X]$ denotes the expectation of a random variable $X$. We use bold lower-case letters for vectors and bold upper-case letters for matrices. For a vector $\mathbf{z} \in \mathbb{R}^n$ with entries $z_1, \cdots, z_n$, $\|\mathbf{z}\|_2 = (\sum_{i=1}^n z_i^2)^{1/2}$ denotes the Euclidean norm of $\mathbf{z}$. Given a matrix $\mathbf{A} \in \mathbb{R}^{n \times d}$, we let $\mathbf{a}_i$ denote the $i$-th row, and $a_{ij}$ denote the entry in the $i$-th row and $j$-th column.

**Importance sampling.** All of the methods studied in this paper solve the active regression problem by collecting a single random sample of rows in $\mathbf{A}$ and corresponding entries in $\mathbf{b}$. We introduce a vector of binary random variables $\boldsymbol{\xi} = \{\xi_1, \cdots, \xi_n\}$, where $\xi_i$ is 1 if $\mathbf{a}_i$ (and thus $b_i$) is selected, and 0 otherwise. $\xi_1, \cdots, \xi_n$ will not necessarily be independent depending on our sampling method. Given a sampling method, let $p_i = \mathbb{E}[\xi_i]$ denote the marginal probability that row $i$ is selected. We return an approximate regression solution as follows: let $\tilde{\mathbf{A}} \in \mathbb{R}^{k \times d}$ contain $\mathbf{a}_i/\sqrt{p_i}$ for all $i$ such that $\xi_i = 1$, and similarly let $\tilde{\mathbf{b}} \in \mathbb{R}^k$ contain $b_i/\sqrt{p_i}$ for the same values of $i$. This scaling ensures that, for any fixed $\mathbf{x}$, $\mathbb{E}\|\tilde{\mathbf{A}}\mathbf{x} - \tilde{\mathbf{b}}\|_2^2 = \|\mathbf{A}\mathbf{x} - \mathbf{b}\|_2^2$. To solve the active regression problem, we return $\tilde{\mathbf{x}}^* = \arg\min_{\mathbf{x} \in \mathbb{R}^d} \|\tilde{\mathbf{A}}\mathbf{x} - \tilde{\mathbf{b}}\|_2^2$. Computing $\tilde{\mathbf{x}}^*$ only requires querying $k$ target values in $\mathbf{b}$.

**Leverage Score Sampling.** We consider methods that choose the marginal probabilities proportional to $\mathbf{A}$'s leverage scores. Specifically, our methods sample row $\mathbf{a}_i$ with marginal probability $\tilde{p}_i = \min(1, c_k \cdot \tau_i)$, where $c_k$ is chosen so that $\sum_{i=1}^n \tilde{p}_i = k$. Details of how to find $c_k$ are discussed in Appendix A. We note that we always have $c_k \geq k/d$ since $\sum_{i=1}^n p_i \leq \sum_{i=1}^n \frac{k}{d} \cdot \tau_i \leq \frac{k}{d} \cdot d = k$.

## 2 OUR METHODS

In this section, we present our sampling scheme which consists of two steps; deterministically constructing a binary tree, and choosing samples by running the pivotal method on this tree. The pivotal method is described in Algorithm 1. It takes as input a binary tree with $n$ leaf nodes, each corresponding to a single index $i$ to be sampled. For each index, we also have an associated probability $\tilde{p}_i$. The algorithm collects a set of exactly $k$ samples $\mathcal{S}$ where $k = \sum_{i=1}^n \tilde{p}_i$. It does so by percolating up the tree and performing repeated head-to-head comparisons of the indices at sibling nodes in the tree. After each comparison, one node promotes to the parent node with updated inclusion probability, and the other node is determined to be sampled or not to be sampled.

It can be checked that, after running Algorithm 1, index $i$ is always sampled with probability $\tilde{p}_i$, regardless of the choice of $T$. However, the samples collected by the pivotal method are not independent, but rather negatively correlated: siblings in $T$ are unlikely to both be sampled, and in general, the events that close neighbors in the tree are both sampled are negatively correlated. In particular, if index $i$ could at some point compete with an index $j$ in the pivotal process, the chance of selecting $j$ decreases if we condition on $i$ being selected. We take advantage of this property to generate spatially distributed samples by constructing a binary tree that matches the underlying geometry of our data. In particular, assume we are given a set of points $\mathbf{X} \in \mathbb{R}^{n \times d'}$. $\mathbf{X}$ will eventually be used to construct a regression matrix $\mathbf{A} \in \mathbb{R}^{n \times d}$ via feature transformation (e.g. by adding polynomial features). However, we construct the sampling tree using $\mathbf{X}$ alone.

---

**Algorithm 1** Binary Tree Based Pivotal Sampling (Deville and Tille, 1998)

---

**Input:** Depth $t$ full binary tree $T$ with $n$ leaves, inclusion probabilities $\{\tilde{p}_1, \cdots, \tilde{p}_n\}$ for each leaf.
**Output:** Set of $k$ sampled indices $\mathcal{S}$.
 1: Initialize $\mathcal{S} = \emptyset$.
 2: **while** $T$ has at least two remaining children nodes **do**
 3:     Select any pair of sibling nodes $S_1, S_2$ with parent $P$. Let $i, j$ be the indices stored at $S_1, S_2$.
 4:     **if** $\tilde{p}_i + \tilde{p}_j \leq 1$ **then**
 5:         With probability $\frac{\tilde{p}_i}{\tilde{p}_i + \tilde{p}_j}$, set $\tilde{p}_i \leftarrow \tilde{p}_i + \tilde{p}_j, \tilde{p}_j \leftarrow 0$. Store $i$ at $P$.
 6:         Otherwise, set $\tilde{p}_j \leftarrow \tilde{p}_i + \tilde{p}_j, \tilde{p}_i \leftarrow 0$. Store $j$ at $P$.
 7:     **else if** $\tilde{p}_i + \tilde{p}_j > 1$ **then**
 8:         With probability $\frac{1 - \tilde{p}_i}{2 - \tilde{p}_i - \tilde{p}_j}$, set $\tilde{p}_i \leftarrow \tilde{p}_i + \tilde{p}_j - 1, \tilde{p}_j = 1$. Store $i$ at $P$ and set $\mathcal{S} \leftarrow \mathcal{S} \cup \{j\}$.
 9:         Otherwise, set $\tilde{p}_j \leftarrow \tilde{p}_i + \tilde{p}_j - 1, \tilde{p}_i \leftarrow 1$. Store $j$ at $P$ and set $\mathcal{S} \leftarrow \mathcal{S} \cup \{i\}$.
10:     Remove $S_1, S_2$ from $T$.
11: **return** $\mathcal{S}$

---

**Algorithm 2** Binary Tree Construction by Coordinate or PCA Splitting

---

**Input:** Matrix $\mathbf{X} \in \mathbb{R}^{n \times d'}$, split method $\in \{$PCA, coordinate$\}$, inclusion probabilities $\tilde{p}_1, \cdots, \tilde{p}_n$.
**Output:** Binary tree $\mathcal{T}$ where each leaf corresponds to a row in $\mathbf{X}$.
 1: Create a tree $T$ with a single root node. Assign set $\mathcal{R}$ to the root where $\mathcal{R} = \{i \in [n]; \tilde{p}_i < 1\}$.
 2: **while** There exists a node in $T$ that holds set $\mathcal{K}$ such that $|\mathcal{K}| > 1$ **do**
 3:     Select any such node $N$ and let $t$ be its level in the tree. Construct $\mathbf{X}_{(\mathcal{K})} \in \mathbb{R}^{|\mathcal{K}| \times d'}$.
 4:     **if** split method $=$ PCA **then**
 5:         Sort $\mathbf{X}_{(\mathcal{K})}$ according to the direction of the maximum variance.
 6:     **else if** split method $=$ coordinate **then**
 7:         Sort $\mathbf{X}_{(\mathcal{K})}$ according to values in its $((t \bmod d') + 1)$-th column.
 8:     Create a left child of $N$. Assign to it all indices associated with the first $\lfloor \frac{|\mathcal{K}|}{2} \rfloor$ rows of $\mathbf{X}_{(\mathcal{K})}$.
 9:     Create a right child of $N$. Assign to it the all remaining indices in $\mathbf{X}_{(\mathcal{K})}$. Delete $\mathcal{K}$ from $N$.
10: **return** $T$

---

Our tree construction method is given as Algorithm 2. $\mathbf{X}_{\mathcal{K}}$ denotes the subset of rows of $\mathbf{X}$ with indices in the set $\mathcal{K}$. First, the algorithm eliminates all data points with inclusion probability $\tilde{p}_i = 1$. Next, it recursively partitions the remaining data points into two subgroups of the same size until all the subgroups have only one data point. Our two methods, PCA-based and coordinate-wise, only differ in how to partition. The PCA-based method performs principal component analysis to find the direction of the maximum variance and splits the space by a hyperplane orthogonal to the direction so that the numbers of data points on both sides are equal. The coordinate-wise version takes a coordinate (corresponding to a column in $\mathbf{X}$) in cyclic order and divides the space by a hyperplane orthogonal to the chosen coordinate. An illustration of the PCA-based binary tree construction run on a fine uniform grid of data points in $\mathbb{R}^2$ is shown in Figure 3. Note that our tree construction method ensures the the number of indices assigned to each subgroup (color) at each level is equal to with $\pm 1$ point. As such, we end of with an even partition of data points into spatially correlated sets. Two indices will be more negatively correlated if they lie in the same set at a higher depth number.

## 3 THEORETICAL ANALYSIS

As will be shown experimentally, when using probabilities $\tilde{p}_1, \ldots, \tilde{p}_n$ proportional to the statistical leverage scores of $\mathbf{A}$, our tree-based pivotal methods significantly outperform its Bernoulli counterpart for active regression. We provide two results theoretically justifying this operation. We first show that, no matter what our original data matrix $\mathbf{X}$ is, and what feature transformation is used to construct $\mathbf{A}$, our methods never perform *worse* than Bernoulli sampling. In particular, they match the $O(d \log d + d/\epsilon)$ sample complexity of independent leverage score sampling. This result is stated as Theorem 1.1. Its proof is given to Appendix B, but we outline our main approach here.

Following existing proofs for independent random sampling (e.g. (Woodruff, 2014) Theorem 1.1 requires two main ingredients: a subspace embedding result, and an approximate matrix-vector multiplication result. In particular, let $\mathbf{U} \in \mathbb{R}^{n \times d}$ be any orthogonal span for the columns of $\mathbf{A}$. Let $\mathbf{S} \in \mathbb{R}^{k \times n}$ be a subsampling matrix that contains a row for every index $i$ selected by our sampling

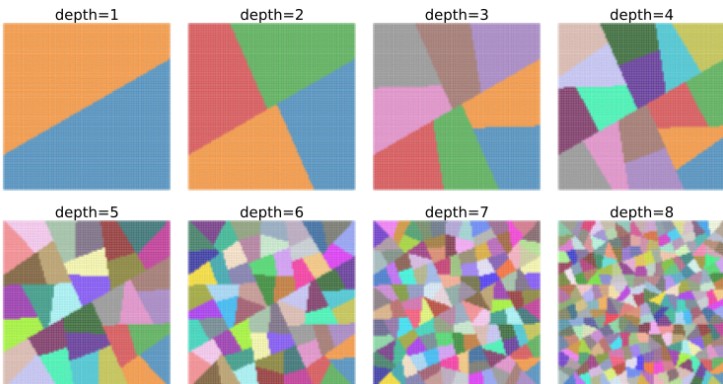

Figure 3: Visualization of a binary tree constructed via our Algorithm 2 using the PCA method for a matrix $\mathbf{X} \in \mathbb{R}^{n \times 2}$ containing points on a uniform square grid. For each depth, data points are given the same color if they compose a subtree with root at that depth. As we can see, the method produces uniform recursive spatial partitions, which encourage spatially separated samples.

scheme, which has value $1/\sqrt{p_i}$ at entry $i$, and is $0$ everywhere else. So, in the notation of Theorem 1.1, $\tilde{\mathbf{A}} = \mathbf{SA}$ and $\tilde{\mathbf{b}} = \mathbf{Sb}$. To prove the theorem, it suffices to show that with high probability,

1. **Subspace Embedding:** For all $\mathbf{x} \in \mathbb{R}^d$, $\frac{1}{2}\|\mathbf{x}\|_2 \leq \|\mathbf{SUx}\|_2 \leq 1.5\|\mathbf{x}\|_2$.

2. **Approximate Matrix-Vector Multiplication**: $\|\mathbf{U}^T\mathbf{S}^T\mathbf{S}(\mathbf{b} - \mathbf{Ax}^*)\|_2^2 \leq \epsilon\|\mathbf{b} - \mathbf{Ax}^*\|_2^2$.

The first property is equivalent to $\|\mathbf{U}^T\mathbf{S}^T\mathbf{SU} - \mathbf{I}\|_2 \leq 1/2$. I.e., after subsampling, $\mathbf{U}$ should remain nearly orthogonal. The second property requires that after subsampling with $\mathbf{S}$, the optimal residual, $\mathbf{b} - \mathbf{Ax}^*$, should have small product with $\mathbf{U}$. Note that without subsampling, $\|\mathbf{U}^T(\mathbf{b} - \mathbf{Ax}^*)\|_2^2 = 0$.

We show that both of the above bounds can be established for any sampling method that 1) samples index $i$ with marginal probability proportional to its leverage score 2) is homogeneous, meaning that it takes a fixed number of samples $k$, and 3) produces a distribution over binary vectors satisfying the following property:

**Definition 3.1** (One-sided $\ell_\infty$-independence). *Let $\xi_1, \cdots, \xi_n \in \{0, 1\}^n$ be random variables with joint distribution $\mu$. Let $\mathcal{S} \subseteq [n]$ and let $i, j \in [n] \backslash \mathcal{S}$. Define the one-sided influence matrix $\mathcal{I}_\mu^\mathcal{S}$ as:*

$$\mathcal{I}_\mu^\mathcal{S}(i, j) = \Pr_\mu[\xi_j = 1 | \xi_i = 1 \wedge \xi_\ell = 1 \forall \ell \in \mathcal{S}] - \Pr_\mu[\xi_j = 1 | \xi_\ell = 1 \forall \ell \in \mathcal{S}]$$

*Let $\|\mathcal{I}_\mu^\mathcal{S}\|_\infty = \max_{i \in [n]} \sum_{j \in [n]} |\mathcal{I}_\mu^\mathcal{S}(i, j)|$. $\mu$ is one-sided $\ell_\infty$-independent with param. $D_{inf}$ if, for all $\mathcal{S} \subset [n]$, $\|\mathcal{I}_\mu^\mathcal{S}\|_\infty \leq D_{inf}$. Note that if $\xi_1, \ldots, \xi_n$ are truly independent, we have $D_{inf} = 1$.*

To prove Theorem 1.1, our required subspace embedding property follows immediately from recently established matrix Chernoff-type bounds for sums of random matrices involving one-sided $\ell_\infty$-independent random variables in (Kaufman et al., 2022). In fact, this work is what inspired us to consider this property, as it is the minimal condition under which such bounds are known to hold. Our main theoretical contribution is thus to prove the matrix-vector multiplication property. We do this by generalizing the approach of (Drineas et al., 2006) (which holds for independent random samples) to any distribution that satisfies one-sided $\ell_\infty$-independence.

### 3.1 IMPROVED BOUNDS FOR POLYNOMIAL REGRESSION

We do not believe that Theorem 1.1 can be strengthened in general, as spatially well-spread samples may not be valuable in all settings. For example, in the case when $\mathbf{A} = \mathbf{X}$, the points in $\mathbf{X}$ live in $d$ dimensional space, and we are only collecting $O(d \log d)$ samples, so intuitively any set of points is well-spread. However, we can show that better spatial distribution does offer *provably* better bounds in some settings where $\mathbf{X}$ is low-dimensional and $\mathbf{A}$ is a high dimensional feature transformation. In particular, consider the case when $\mathbf{X}$ is a (quasi) matrix containing just one column, with an infinite number of rows corresponding to every point $t$ in an interval $[\ell, u]$. Every row of $\mathbf{A}$ is a polynomial feature transformation, meaning of the form $\mathbf{a}_t = [1, t, t^2, \ldots, t^d]$ for degree $d$. Now, consider a target vector $\mathbf{b}$ that can be indexed by real numbers in $[\ell, u]$. Solving the infinite dimensional

regression problem $\min_{\mathbf{x}} \|\mathbf{A}\mathbf{x} - \mathbf{b}\|_2^2 = \min_{\mathbf{x}} \int_\ell^u (\mathbf{a}_t^T \mathbf{x} - \mathbf{b}_t)^2 dt$ is equivalent to finding the best polynomial approximation to $\mathbf{b}$ in the $\ell_2$ norm, a well studied application of leverage score sampling Cohen and Migliorati (2017). For this problem, we show that pivotal sampling obtains a sample complexity of $O(d/\epsilon)$, improving on independent leverage score sampling by a $\log d$ factor.

This result is stated as Theorem 1.2 and proven in Appendix C. Importantly, we note that the dependence on $d \log d$ in the general active regression analysis comes from the proof of the subspace embedding guarantee – the required approximate matrix-multiplication guarantee already follows with $O(d/\epsilon)$ samples. Our proof eliminates the $\log d$ by avoiding the use of a matrix Chernoff bound entirely. Instead, by taking advantage of connections between leverage scores of the polynomial regression problem and the orthogonality measure of the Chebyshev polynomials, we directly use tools from polynomial approximation theory to prove a subspace embedding bound that holds *deterministically* with just $O(d)$ samples. Our approach is similar to a recently result of Kane et al. (2017), which also obtains $O(d/\epsilon)$ sample complexity for active degree-$d$ polynomial regression, albeit with a sampling method not based on leverage scores.

## 4 EXPERIMENTS

We experimentally evaluate our pivotal sampling methods on active regression problems with low-dimensional structure. The benefits of leverage score sampling over uniform sampling for such problems has already been established (see e.g. (Hampton and Doostan, 2015) or (Gajjar et al., 2023)) and we provide additional evidence in Appendix D. So, we focus on comparing our pivotal methods to the widely used baseline of *independent* Bernoulli leverage score sampling.

**Test Problems.** We consider several test problems inspired by applications to parametric PDEs. In these problems, we are given a differential equation with parameters, and seek to compute some observable quantity of interest (QoI) for different choices of parameters. This can be done directly by solving the PDE, but doing so is computationally expensive. Instead, the goal is to use a small number of solutions to the PDE to fit a surrogate model (in our case, a low-degree polynomial) that approximates the QoI well, either over a given parameter range, or on average for parameters drawn from a distribution (e.g., Gaussian). The problem is naturally an active regression problem because we can choose exactly what parameters to solve the PDE for. The first equation we consider models the displacement $x$ of a damped harmonic oscillator with a sinusoidal force over time $t$.

$$\frac{d^2x}{dt^2}(t) + c\frac{dx}{dt}(t) + kx(t) = f\cos(\omega t), \quad x(0) = x_0, \quad \frac{dx}{dt}(0) = x_1.$$

The equation has four parameters; damping coefficient $c$, spring constant $k$, forcing amplitude $f$, and frequency $\omega$. As a QoI, we consider the maximum oscillator displacement after 20 seconds. We fix $c, f = 0.5$, and seek to approximate this displacement over the range domain $k \times \omega = [1, 3] \times [0, 2]$.

We also consider the heat equation for values of $x \in [0, 1]$ with a time-dependent boundary equation and sinusoidal initial condition parameterized by a frequency $\omega$. The heat equation that we consider describes the temperature $f(x, t)$ by the partial differential equation.

$$\pi\frac{\partial f}{\partial t} = \frac{\partial^2 f}{\partial x^2}, \quad f(0, t) = 0, \quad f(x, 0) = \sin(\omega\pi x), \quad \pi e^{-t} + \frac{\partial f(1, t)}{\partial t} = 0.$$

(a) Damped Harmonic Oscillator, degree $p = 12$. (b) Heat Equation, degree $p = 12$. (c) Surface Reaction, degree $p = 15$. (d) Surface Reaction, degree $p = 20$.

Figure 4: Results for active polynomial regression for the damped harmonic oscillator QoI, the heat equation QoI, and the surface reaction model with polynomials varying degree. Our leverage-score based pivotal method outperforms standard Bernoulli leverage score sampling, suggesting the benefits of spatially-aware sampling.

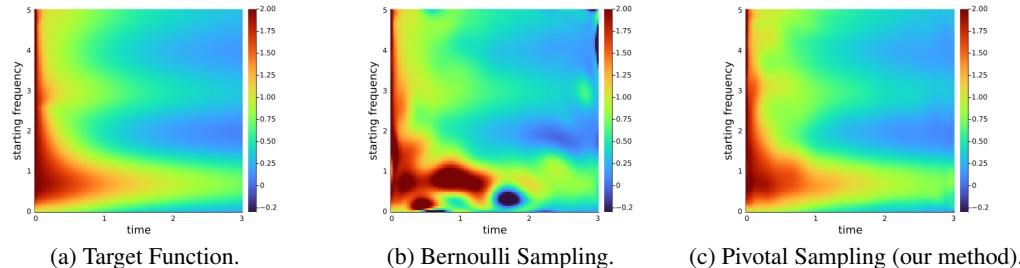

(a) Target Function.  (b) Bernoulli Sampling.  (c) Pivotal Sampling (our method).

Figure 5: Polynomial approximation to the maximum temperature of a heat diffusion problem, as a function of time and starting condition. (a) is the target value and both (b) and (c) draw 240 samples using the leverage score and perform polynomial regression of degree 20. However, (b) uses Bernoulli sampling while (c) employs our PCA-based pivotal sampling.

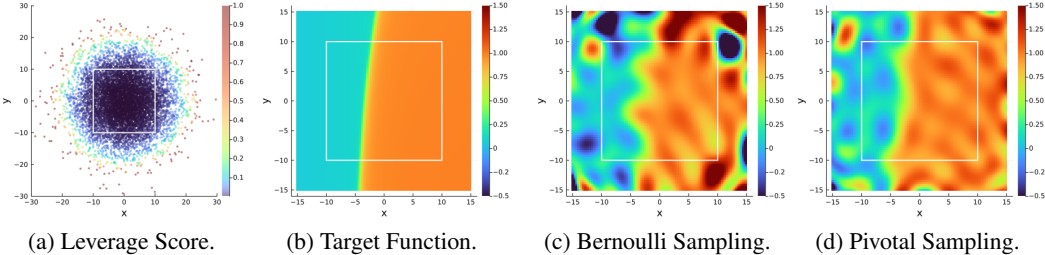

(a) Leverage Score.  (b) Target Function.  (c) Bernoulli Sampling.  (d) Pivotal Sampling.

Figure 6: Approximation of the surface reaction model when $425$ samples are used to fit a degree $25$ polynomial. To help readers focus on the area near the origin, we draw $[-10, 10]^2$ box.

As a QoI, we estimate the maximum temperature over all values of $x$ for $t \in [0, 3]$ and $\omega \in [0, 5]$.

**Data Matrix.** For both problems, we construct $\mathbf{A}$ by uniformly selecting $n = 10^5$ data points in the 2-dimensional parameter range of interest. We then add all polynomial features of degree $p = 12$ as discussed in Section 1.2. We compute sampled entries from the target vector $\mathbf{b}$ using standard MATLAB routines. Results comparing our PCA-based pivotal method and Bernoulli leverage score sampling are show in Figure 4. We report median normalized error $\|\mathbf{A}\tilde{\mathbf{x}}^* - \mathbf{b}\|_2^2 / \|\mathbf{b}\|_2^2$ after $1000$ trials. By drawing more samples from $\mathbf{b}$, the errors of all methods eventually converge to the optimal error $\|\mathbf{A}\mathbf{x}^* - \mathbf{b}\|_2^2 / \|\mathbf{b}\|_2^2$, but clearly the pivotal method requires less samples to achieve a given level of accuracy, confirming the benefits of spatially-aware sampling. We also visualize results for the damped harmonic oscillator in Figure 1, showing approximations obtained with 250 samples. Visualizations for the heat equation are given in Figure 5. For both targets, one can directly see that pivotal sampling improves the performance over Bernoulli sampling.

We also consider a chemical surface coverage problem from (Hampton and Doostan, 2015). Details are relegated to Appendix D. Again, we vary two parameters and seek to fit a surrogate model using a small number of example pairs of parameters. Instead of a uniform distribution, for this problem, the rows of $\mathbf{X}$ are drawn from a Gaussian distribution with 0-mean and 7.5 standard deviation. We construct $\mathbf{A}$ using polynomial features of varying degrees. Convergence results are shown in Figure 4 and the fit visualized for data near the origin in Figure 6. This is a challenging problem since the target function has sharp threshold behavior that is difficult to approximate with a polynomial. However, our pivotal based leverage score sampling performs well, providing a better fit than Bernoulli sampling. Additional experiments, including on 3D problems are reported in Appendix D.

## 5  CONCLUSION AND FUTURE WORK

In this paper, we introduce a *spatially-aware* pivotal sampling method and empirically demonstrate its effectiveness for active linear regression for spatially-structured problems. We prove a general theorem that can be used to analyze the number of samples required in the agnostic setting for any similar method that samples with marginal probabilities proportional to the leverage scores using a distribution that satisfies one-sided $\ell_\infty$ independence. We provide a stronger bound for the important special case of polynomial regression, showing that our method can obtain a sample complexity of $O(d/\epsilon)$, removing a $\log(d)$ factor from independent leverage score sampling. This result provides initial theoretical evidence for the strong performance of pivotal sampling in practice. Extending it to other spatially structured function classes would be an interesting direction for future work.

ACKNOWLEDGEMENTS

Xiaoou Cheng and Jonathan Weare were supported by NSF award 2054306. Christopher Musco was supported by DOE award DE-SC0022266 and NSF award 2045590. We would like to thank NYU IT for the use of the Greene computing cluster.

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

## A   PROBABILITY PRE-PROCESSING

As discussed in Section 1.4, our sampling methods require computing probabilities $\tilde{p}_1, \ldots, \tilde{p}_n$ where $\tilde{p}_i = \min(1, c_k \cdot \tau_i)$ for some fixed constant $c_k$ chosen so that $\sum_{i=1}^{n} \tilde{p}_i = k$. We can find such probabilities using a simple iterative method, which takes as input initial "probabilities" $p_i = \frac{k}{d}\tau_i$ that are proportional to the leverage scores $\tau_1, \ldots, \tau_n$ of $\mathbf{A}$. Note that $p_i$ could be larger than 1 if $k > d$. Pseudocode for how to adjust these probabilities is included in Algorithm 3.

---

**Algorithm 3** Probability Ceiling

**Input:** Number of samples to choose $k$, inclusion probabilities $\{p_1, \cdots, p_n\}$.
 1: Set $\tilde{p}_i = p_i$ for all $i \in [n]$.
 2: **while** $\mathcal{S} = \{i \in [n]; \tilde{p}_i > 1\}$ is not empty **do**
 3:     $\tilde{p}_i \leftarrow 1$ for all $i \in \mathcal{S}$.
 4:     $\mathcal{D} = \{i \in [n]; \tilde{p}_i = 1\}$, $\mathcal{R} = \{i \in [n]; \tilde{p}_i < 1\}$.
 5:     $\tilde{p}_i \leftarrow \frac{k-|\mathcal{D}|}{\sum_{i \in \mathcal{R}} \tilde{p}_i} \tilde{p}_i$ for all $i \in \mathcal{R}$.
 6: **return** $\{\tilde{p}_1, \cdots, \tilde{p}_n\}$

---

To see that the method returns $\tilde{p}_i$ with the desired properties, first note that $\frac{k-|\mathcal{D}|}{\sum_{i \in \mathcal{R}} \tilde{p}_i}$ is always greater than 1. In particular, at the beginning of the while loop, we always have the invariant that $\sum_{i=1}^{n} \tilde{p}_i = k$. Accordingly, $\sum_{i \in \mathcal{R}} \tilde{p}_i = k - \sum_{i \in \mathcal{D}} \tilde{p}_i < k - |\mathcal{D}|$. As a result, at the end of the algorithm, $\tilde{p}_i$ certainly equals $\min(1, c \cdot \tau_i)$ for some constant $c \geq 1$. And in fact, it must be that $c = c_k$ by the invariant that $\sum_{i=1}^{n} \tilde{p}_i = k$.

The main loop in Algorithm 3 terminates as soon as $\mathcal{S}$ is empty, which happens after at most $k$ steps. In practice however, the method usually converges in 2 or 3 iterations. Nevertheless, our primary objective is to minimize the number of samples required for an accurate regression solution – we do not consider runtime costs in detail.

## B   PROOF OF THEOREM 1.1

In this section, we prove our first theoretical result, which holds for any row sampling distribution whose marginal probabilities are proportional to the leverage scores of $\mathbf{A}$, and which satisfies a few additional conditions. In particular, we require that the distribution is both $k$-homogeneous and is $\ell_\infty$-independent with constant parameter $D$. We define the first requirement below, and recall the definition of $\ell_\infty$-independence from Section 3. For both definitions, we view our subsampling process as a method that randomly selects a binary vector $\boldsymbol{\xi} = \{\xi_1, \cdots, \xi_n\}$ from $\{0, 1\}^n$. Entries of 1 in the vector correspond to rows that are sampled from $\mathbf{A}$ (and thus labels that are observed

in b). Under this notation, our requirement on marginal probabilities is equivalent to requiring that $\mathbb{E}[\xi_i] = \min(1, c \cdot \tau_i) = \tilde{p}_i$ for some fixed constant $c$, where $\tau_i$ is the $i$-th leverage score of $\mathbf{A}$.

**Definition B.1** (Homogeneity). *A distribution over binary vectors $\boldsymbol{\xi} = \{\xi_1, \cdots, \xi_n\}$ is $k$-homogeneous if all possible realizations of $\boldsymbol{\xi}$ contain exactly $k$ ones.*

**Definition 3.1** (One-sided $\ell_\infty$-independence). *Let $\xi_1, \cdots, \xi_n \in \{0,1\}^n$ be random variables with joint distribution $\mu$. Let $\mathcal{S} \subseteq [n]$ and let $i, j \in [n] \backslash \mathcal{S}$. Define the one-sided influence matrix $\mathcal{I}_\mu^{\mathcal{S}}$ as:*

$$\mathcal{I}_\mu^{\mathcal{S}}(i,j) = \Pr_\mu[\xi_j = 1 | \xi_i = 1 \wedge \xi_\ell = 1 \forall \ell \in \mathcal{S}] - \Pr_\mu[\xi_j = 1 | \xi_\ell = 1 \forall \ell \in \mathcal{S}]$$

*Let $\|\mathcal{I}_\mu^{\mathcal{S}}\|_\infty = \max_{i \in [n]} \sum_{j \in [n]} |\mathcal{I}_\mu^{\mathcal{S}}(i,j)|$. $\mu$ is one-sided $\ell_\infty$-independent with param. $D_{inf}$ if, for all $\mathcal{S} \subset [n]$, $\|\mathcal{I}_\mu^{\mathcal{S}}\|_\infty \leq D_{inf}$. Note that if $\xi_1, \ldots, \xi_n$ are truly independent, we have $D_{inf} = 1$.*

With these definitions in place, we are ready for our proof:

*Proof of Theorem 1.1.* As outlined in Section 3, we need to establish two results: a subspace embedding guarantee and an approximate matrix-vector multiplication result. We prove these separately below, then combine the results to complete the proof of the theorem.

**Subspace Embedding.** Let $\mathbf{S}$ be a subsampling matrix corresponding to a vector $\boldsymbol{\xi}$ selected from a distribution with the properties above. That is, $\mathbf{S} \in \mathbb{R}^{k \times n}$ contains a row for every index $i$ where $\xi_i = 1$. That row has value $1/\sqrt{\tilde{p}_i}$ at entry $i$, and is 0 everywhere else. Let $\mathbf{U} \in \mathbb{R}^{n \times d}$ be an orthonormal span for $\mathbf{A}$'s columns. We will show that, if $k = O\left(\frac{d \log(d/\delta)}{\alpha^2} \cdot D_{inf}^2\right)$, with prob. $1 - \delta$,

$$\|\mathbf{U}^T \mathbf{S}^T \mathbf{S} \mathbf{U} - \mathbf{I}\|_2 \leq \alpha. \tag{B.1}$$

Note that, since $\left|\|\mathbf{S}\mathbf{x}\|_2^2 - \|\mathbf{U}\mathbf{x}\|_2^2\right| = \left|\mathbf{x}^T \left(\mathbf{U}^T \mathbf{S}^T \mathbf{S} \mathbf{U} - \mathbf{I}\right) \mathbf{x}\right| \leq \|\mathbf{U}^T \mathbf{S}^T \mathbf{S} \mathbf{U} - \mathbf{I}\|_2 \|\mathbf{x}\|_2^2$, this is equivalent to showing that, for all $\mathbf{x} \in \mathbb{R}^d$,

$$(1 - \alpha)\|\mathbf{U}\mathbf{x}\|_2^2 \leq \|\mathbf{S}\mathbf{x}\|_2^2 \leq (1 + \alpha)\|\mathbf{U}\mathbf{x}\|_2^2.$$

We establish (B.1) using the following matrix Chernoff bound from (Kaufman et al., 2022):

**Lemma B.1** (Matrix Chernoff for $\ell_\infty$-independent Distributions). *Let $\xi_1, \cdots, \xi_n$ be binary random variables with joint distribution $\mu$ that is $z$-homogeneous for any positive integer $z$ and $\ell_\infty$-independent with parameter $D_{inf}$. Let $\mathbf{Y}_1, \ldots, \mathbf{Y}_n \in \mathbb{R}^{d \times d}$ be positive semidefinite (PSD) matrices with spectral norm $\|\mathbf{Y}_i\|_2 \leq R$ for some $R > 0$. Let $\mu_{\max} = \lambda_{\max}(\mathbb{E}_{\boldsymbol{\xi} \sim \mu}[\sum_{i=1}^n \xi_i \mathbf{Y}_i])$ and $\mu_{\min} = \lambda_{\min}(\mathbb{E}_{\boldsymbol{\xi} \sim \mu}[\sum_{i=1}^n \xi_i \mathbf{Y}_i])$. Then, for any $\alpha \in (0,1)$ and a fixed constant $c$,*

$$\Pr\left[\lambda_{\max}\left(\sum_{i=1}^n \xi_i \mathbf{Y}_i\right) \geq (1 + \alpha)\mu_{\max}\right] \leq d \exp\left(-\frac{\alpha^2 \mu_{\max}}{cR D_{inf}^2}\right),$$

$$\Pr\left[\lambda_{\min}\left(\sum_{i=1}^n \xi_i \mathbf{Y}_i\right) \leq (1 - \alpha)\mu_{\min}\right] \leq d \exp\left(-\frac{\alpha^2 \mu_{\min}}{cR D_{inf}^2}\right).$$

Let $\mathcal{D} = \{i \in [n] : \tilde{p}_i = 1\}$ and $\mathcal{R} = \{i \in [n] : \tilde{p}_i < 1\}$. Letting $\mathbf{u}_i$ denote the $i$-th row of $\mathbf{U}$, $\mathbf{U}^T \mathbf{S}^T \mathbf{S} \mathbf{U}$ can be decomposed as:

$$\mathbf{U}^T \mathbf{S}^T \mathbf{S} \mathbf{U} = \sum_{i \in \mathcal{R}} \xi_i \frac{\mathbf{u}_i \mathbf{u}_i^T}{\tilde{p}_i} + \sum_{i \in \mathcal{D}} \mathbf{u}_i \mathbf{u}_i^T.$$

Recall that $\tilde{p}_i = \min(1, c \cdot \tau_i)$ for some constant $c$, and in fact, since $\sum_{i=1}^n \tilde{p}_i = k$ and $\sum_{i=1}^n \tau_i \leq d$, it must be that $c \geq \frac{k}{d}$. Define $p_i = \frac{k}{d}\tau_i$. For each $i \in \mathcal{D}$, let $m_i$ be an integer such that $m_i \geq p_i$. For $i \in \mathcal{D}$, $p_i = \frac{k}{d}\tau_i \geq 1$, so $m_i \geq 1$. Additionally let $\xi_i^1, \ldots, \xi_i^{m_i} = \xi_i$ be variables that are deterministically 1. Then we can trivially split up the second sum above so that:

$$\mathbf{U}^T \mathbf{S}^T \mathbf{S} \mathbf{U} = \sum_{i \in \mathcal{R}} \xi_i \frac{\mathbf{u}_i \mathbf{u}_i^T}{\tilde{p}_i} + \sum_{i \in \mathcal{D}} \sum_{j=1}^{m_i} \xi_i^j \frac{\mathbf{u}_i \mathbf{u}_i^T}{m_i}. \tag{B.2}$$

It is easy to see that $\{\xi_i : i \in \mathcal{R}\} \cup \{\xi_i^j : i \in \mathcal{D}, j \in [m_i]\}$ are $\ell_\infty$-independent random variables with parameter $D_{\inf}$ and form a distribution that is $\bar{k}$ homogeneous for $\bar{k} = k + \sum_{i \in \mathcal{D}}(m_i - 1)$. So, noting that $\mathbf{u}_i \mathbf{u}_i^T$ is PSD, we can apply Lemma B.1 to (B.2). We are left to bound $R$ and $\mu_{\max}$.

To bound $R$, for $i \in \mathcal{R}$, let $\mathbf{Y}_i = \frac{\mathbf{u}_i \mathbf{u}_i^T}{\tilde{p}_i}$. For such $i$, we have that $\tilde{p}_i \geq p_i$. So,

$$\|\mathbf{Y}_i\|_2 = \frac{\|\mathbf{u}_i \mathbf{u}_i^T\|_2}{\tilde{p}_i} = \frac{\|\mathbf{u}_i\|_2^2}{\tilde{p}_i} = \frac{\tau_i}{\tilde{p}_i} \leq \frac{\tau_i}{p_i} = \frac{d}{k}.$$

I.e., for all $i \in \mathcal{R}$, $\|\mathbf{Y}_i\|_2 \leq R$ for $R = \frac{d}{k}$. Additionally, for $i \in \mathcal{D}$, let $\mathbf{Y}_i = \frac{\mathbf{u}_i \mathbf{u}_i^T}{m_i}$. Since we chose $m_i \geq p_i$, by the same argument, we have that $\|\mathbf{Y}_i\|_2 \leq R$ for $R = \frac{d}{k}$.

Next, we bound $\mu_{\max}$ by simply noting that

$$\mathbb{E}\left[\mathbf{U}^T \mathbf{S}^T \mathbf{S} \mathbf{U}\right] = \sum_{i \in \mathcal{R}} \mathbb{E}[\xi_i] \frac{\mathbf{u}_i \mathbf{u}_i^T}{\tilde{p}_i} + \sum_{i \in \mathcal{D}} \mathbf{u}_i \mathbf{u}_i^T = \sum_{i \in \mathcal{R}} \mathbf{u}_i \mathbf{u}_i^T + \sum_{i \in \mathcal{D}} \mathbf{u}_i \mathbf{u}_i^T = \mathbf{I},$$

where $\mathbf{I}$ denotes the $d \times d$ identity matrix. It follows that $\mu_{\max} = \mu_{\min} = 1$.

Plugging into Lemma B.1 and applying a union bound, we obtain

$$\Pr\left[\|\mathbf{U}^T \mathbf{S}^T \mathbf{S} \mathbf{U} - \mathbf{I}\|_2 \geq \alpha\right] \leq d \exp\left(-\frac{k\alpha^2}{c' D_{\inf}^2 d}\right),$$

where $c'$ is a fixed constant. Setting $k = O(\frac{d \log(d/\delta)}{\alpha^2} \cdot D_{\inf}^2)$, we have $\|\mathbf{U}^T \mathbf{S}^T \mathbf{S} \mathbf{U} - \mathbf{I}\|_2 \leq \alpha$ with probability at least $1 - \delta$. This establishes (B.1).

**Approximate Matrix-Vector Multiplication**. With our subspace embedding result in place, we move onto proving the necessary approximate matrix-vector multiplication result introduced in Section 3. In particular, we wish to show that, with good probability, $\|\mathbf{U}^T \mathbf{S}^T \mathbf{S}(\mathbf{b} - \mathbf{A}\mathbf{x}^*)\|_2^2 \leq \epsilon\|\mathbf{b} - \mathbf{A}\mathbf{x}^*\|_2^2$ where $\mathbf{x}^* = \arg\min_{\mathbf{x}} \|\mathbf{A}\mathbf{x} - \mathbf{b}\|_2^2$. Since the residual of the optimal solution, $\mathbf{b} - \mathbf{A}\mathbf{x}^*$, is the same under any column transformation of $\mathbf{A}$, we have $\mathbf{b} - \mathbf{A}\mathbf{x}^* = \mathbf{b} - \mathbf{U}\mathbf{y}^*$ where $\mathbf{y}^* = \arg\min_{\mathbf{y}} \|\mathbf{U}\mathbf{y} - \mathbf{b}\|_2^2$. We can then equivalently show that, if we take $k = O\left(\frac{d}{\epsilon\delta} \cdot D_{\inf}\right)$ samples, with probability $1 - \delta$,

$$\|\mathbf{U}^T \mathbf{S}^T \mathbf{S}(\mathbf{b} - \mathbf{U}\mathbf{y}^*)\|_2^2 \leq \epsilon\|\mathbf{b} - \mathbf{U}\mathbf{y}^*\|_2^2. \tag{B.3}$$

As in the proof for independent random samples (Drineas et al., 2006), we will prove (B.3) by bounding the expected squared error and applying Markov's inequality. In particular, we have

$$\Pr\left[\|\mathbf{U}^T \mathbf{S}^T \mathbf{S}(\mathbf{b} - \mathbf{U}\mathbf{y}^*)\|_2^2 \geq \epsilon\|\mathbf{b} - \mathbf{U}\mathbf{y}^*\|_2^2\right] \leq \frac{\mathbb{E}\left[\|\mathbf{U}^T \mathbf{S}^T \mathbf{S}(\mathbf{b} - \mathbf{U}\mathbf{y}^*)\|_2^2\right]}{\epsilon\|\mathbf{b} - \mathbf{U}\mathbf{y}^*\|_2^2}. \tag{B.4}$$

Let $\mathbf{z} = \mathbf{b} - \mathbf{U}\mathbf{y}^*$ and note that $\mathbf{U}^T \mathbf{z} = \mathbf{0}$. The numerator on the right side can be transformed as

$$\mathbb{E}\left[\|\mathbf{U}^T \mathbf{S}^T \mathbf{S}\mathbf{z}\|_2^2\right] = \mathbb{E}\left[\|\mathbf{U}^T \mathbf{S}^T \mathbf{S}\mathbf{z} - \mathbf{U}^T \mathbf{z}\|_2^2\right] = \mathbb{E}\left[\|\mathbf{U}^T (\mathbf{S}^T \mathbf{S} - \mathbf{I})\mathbf{z}\|_2^2\right].$$

Note that above $\mathbf{S}^T \mathbf{S} - \mathbf{I}$ is a diagonal matrix with $i$-th diagonal entry equal to $\frac{1}{\tilde{p}_i} - 1$ if $\xi_i = 1$ and $-1$ if $\xi_i = 0$. Expanding the $\ell_2$-norm and using that $\xi_i = 1$ and $\tilde{p}_i = 1$ for $i \notin \mathcal{R}$, we have

$$\mathbb{E}\left[\|\mathbf{U}^T \mathbf{S}^T \mathbf{S}\mathbf{z}\|_2^2\right] = \sum_{j=1}^{d} \mathbb{E}\left[\left(\sum_{i=1}^{n} \left(\frac{\xi_i}{\tilde{p}_i} - 1\right) u_{ij} z_i\right)^2\right] = \sum_{j=1}^{d} \mathbb{E}\left[\left(\sum_{i \in \mathcal{R}} \left(\frac{\xi_i}{\tilde{p}_i} - 1\right) u_{ij} z_i\right)^2\right]$$

$$= \sum_{j=1}^{d} \sum_{i \in \mathcal{R}} \sum_{l \in \mathcal{R}} \frac{c_{il}}{\tilde{p}_i \tilde{p}_l} u_{ij} z_i u_{lj} z_l,$$

where $c_{il} = \text{Cov}(\xi_i, \xi_l) = \mathbb{E}[(\xi_i - \tilde{p}_i)(\xi_l - \tilde{p}_l)] = \mathbb{E}[\xi_i \xi_l] - \tilde{p}_i \tilde{p}_l$. We will show that

$$\sum_{j=1}^{d} \sum_{i \in \mathcal{R}} \sum_{l \in \mathcal{R}} \frac{c_{il}}{\tilde{p}_i \tilde{p}_l} u_{ij} z_i u_{lj} z_l \leq D_{\inf} \sum_{j=1}^{d} \sum_{i \in \mathcal{R}} \frac{u_{ij}^2 z_i^2}{\tilde{p}_i},$$

where $D_{\text{inf}}$ is the $\ell_\infty$-independence parameter. It suffices to show that for every $j$, we have

$$\sum_{i \in \mathcal{R}} \sum_{l \in \mathcal{R}} \frac{c_{il}}{\tilde{p}_i \tilde{p}_l} u_{ij} z_i u_{lj} z_l \le D_{\text{inf}} \sum_{i \in \mathcal{R}} \frac{u_{ij}^2 z_i^2}{\tilde{p}_i}. \tag{B.5}$$

Define a symmetric matrix $\mathbf{M} \in \mathbb{R}^{|\mathcal{R}| \times |\mathcal{R}|}$ with entries $m_{il} = \frac{c_{il}}{\sqrt{\tilde{p}_i}\sqrt{\tilde{p}_l}}$ and a vector $\mathbf{v} \in \mathbb{R}^{|\mathcal{R}|}$ with entries $v_i = \frac{u_{ij} z_i}{\sqrt{\tilde{p}_i}}$. The desired result in (B.5) can be expressed as

$$\mathbf{v}^T \mathbf{M} \mathbf{v} \le D_{\text{inf}} \|\mathbf{v}\|_2^2.$$

With $\mathcal{S} = \emptyset$, the one-sided $\ell_\infty$-independence condition implies that, for all $i \in [n]$,

$$\sum_{l \in \mathcal{R}} \left| \frac{\mathbb{E}[\xi_i \xi_l]}{\tilde{p}_i} - \tilde{p}_l \right| = \sum_{l \in \mathcal{R}} \frac{\sqrt{\tilde{p}_l}}{\sqrt{\tilde{p}_i}} |m_{il}| \le D_{\text{inf}}.$$

Equivalently, if we define a diagonal matrix $\mathbf{\Lambda} \in \mathbb{R}^{|\mathcal{R}| \times |\mathcal{R}|}$ such that $\Lambda_{ii} = \sqrt{\tilde{p}_i}$, we have shown:

$$\|\mathbf{\Lambda}^{-1} \mathbf{M} \mathbf{\Lambda}\|_\infty \le D_{\text{inf}},$$

where for a matrix $\mathbf{B}$, $\|\mathbf{B}\|_\infty$ denotes $\max_i \sum_l |\mathbf{B}_{il}| = \max_{\mathbf{x}} \frac{\|\mathbf{B}\mathbf{x}\|_\infty}{\|\mathbf{x}\|_\infty}$. It follows that the largest eigenvalue of $\mathbf{\Lambda}^{-1} \mathbf{M} \mathbf{\Lambda}$ is at most $D_{\text{inf}}$ and thus, the largest eigenvalue of $\mathbf{M}$ is also at most $D_{\text{inf}}$. Therefore, we have $\mathbf{v}^T \mathbf{M} \mathbf{v} \le D_{\text{inf}} \|\mathbf{v}\|_2^2$. Considering that $\tilde{p}_i \ge p_i = \frac{k}{d} \tau_i$ for $i \in \mathcal{R}$, we have

$$\sum_{j=1}^d \mathbb{E}\left[ \left( \sum_{i \in \mathcal{R}} \left( \frac{\xi_i}{\tilde{p}_i} - 1 \right) u_{ij} z_i \right)^2 \right] \le D_{\text{inf}} \sum_{j=1}^d \sum_{i \in \mathcal{R}} \frac{u_{ij}^2 z_i^2}{\tilde{p}_i} = D_{\text{inf}} \sum_{i \in \mathcal{R}} \frac{z_i^2}{\tilde{p}_i} \sum_{j=1}^d u_{ij}^2 = D_{\text{inf}} \sum_{i \in \mathcal{R}} \frac{z_i^2}{\tilde{p}_i} \tau_i$$

$$\le D_{\text{inf}} \frac{d}{k} \|\mathbf{z}\|_2^2.$$

Recalling that $\mathbf{z} = \mathbf{U}\mathbf{y}^* - \mathbf{b}$, we can plug into (B.4) with $k = O\left( \frac{d}{\epsilon \delta} \cdot D_{\text{inf}} \right)$ samples, which proves that (B.3) holds with probability at least $1 - \delta$.

**Putting it all together.** With (B.1) and (B.3) in place, we can prove our main result, (1.3) of Theorem 1.1. We follow a similar approach to (Woodruff, 2014). By reparameterization, proving this inequality is equivalent to showing that

$$\|\mathbf{U}\tilde{\mathbf{y}}^* - \mathbf{b}\|_2^2 \le (1 + \epsilon) \|\mathbf{U}\mathbf{y}^* - \mathbf{b}\|_2^2, \tag{B.6}$$

where $\tilde{\mathbf{y}}^* = \arg\min_{\mathbf{y}} \|\mathbf{S}\mathbf{U}\mathbf{y} - \mathbf{S}\mathbf{b}\|_2^2$ and $\mathbf{y}^* = \arg\min_{\mathbf{y}} \|\mathbf{U}\mathbf{y} - \mathbf{b}\|_2^2$. Since $\mathbf{y}^*$ is the minimizer of $\|\mathbf{U}\mathbf{y} - \mathbf{b}\|_2^2$, we have $\nabla_{\mathbf{y}} \|\mathbf{U}\mathbf{y} - \mathbf{b}\|_2^2 = 2\mathbf{U}^T(\mathbf{U}\mathbf{y} - \mathbf{b}) = \mathbf{0}$ at $\mathbf{y}^*$. This indicates that $\mathbf{U}\mathbf{y}^* - \mathbf{b}$ is orthogonal to any vector in the column span of $\mathbf{U}$. Particularly, $\mathbf{U}\mathbf{y}^* - \mathbf{b}$ is orthogonal to $\mathbf{U}\tilde{\mathbf{y}}^* - \mathbf{U}\mathbf{y}^*$. Therefore, by the Pythagorean theorem, we have

$$\|\mathbf{U}\tilde{\mathbf{y}}^* - \mathbf{b}\|_2^2 = \|\mathbf{U}\mathbf{y}^* - \mathbf{b}\|_2^2 + \|\mathbf{U}\tilde{\mathbf{y}}^* - \mathbf{U}\mathbf{y}^*\|_2^2 = \|\mathbf{U}\mathbf{y}^* - \mathbf{b}\|_2^2 + \|\tilde{\mathbf{y}}^* - \mathbf{y}^*\|_2^2. \tag{B.7}$$

So to prove (B.6), it suffices to show that

$$\|\tilde{\mathbf{y}}^* - \mathbf{y}^*\|_2^2 \le \epsilon \|\mathbf{U}\mathbf{y}^* - \mathbf{b}\|_2^2.$$

Applying (B.1) with $k = O(d \log d \cdot D_{\text{inf}}^2 + d \cdot D_{\text{inf}}/\epsilon)$ and $\delta = 1/200$, we have that, with probability $99.5/100$, $\|\mathbf{U}^T \mathbf{S}^T \mathbf{S} \mathbf{U} - \mathbf{I}\|_2 \le \frac{1}{2}$. Then, by triangle inequality,

$$\|\tilde{\mathbf{y}}^* - \mathbf{y}^*\|_2 \le \|\mathbf{U}^T \mathbf{S}^T \mathbf{S} \mathbf{U}(\tilde{\mathbf{y}}^* - \mathbf{y}^*)\|_2 + \|\mathbf{U}^T \mathbf{S}^T \mathbf{S} \mathbf{U}(\tilde{\mathbf{y}}^* - \mathbf{y}^*) - (\tilde{\mathbf{y}}^* - \mathbf{y}^*)\|_2$$

$$\le \|\mathbf{U}^T \mathbf{S}^T \mathbf{S} \mathbf{U}(\tilde{\mathbf{y}}^* - \mathbf{y}^*)\|_2 + \|\mathbf{U}^T \mathbf{S}^T \mathbf{S} \mathbf{U} - \mathbf{I}\|_2 \|\tilde{\mathbf{y}}^* - \mathbf{y}^*\|_2$$

$$\le \|\mathbf{U}^T \mathbf{S}^T \mathbf{S} \mathbf{U}(\tilde{\mathbf{y}}^* - \mathbf{y}^*)\|_2 + \frac{1}{2} \|\tilde{\mathbf{y}}^* - \mathbf{y}^*\|_2. \tag{B.8}$$

Rearranging, we conclude that $\|\tilde{\mathbf{y}}^* - \mathbf{y}^*\|_2^2 \le 4\|\mathbf{U}^T \mathbf{S}^T \mathbf{S} \mathbf{U}(\tilde{\mathbf{y}}^* - \mathbf{y}^*)\|_2^2$. Since $\tilde{\mathbf{y}}^*$ is the minimizer of $\|\mathbf{S}\mathbf{U}\mathbf{y} - \mathbf{S}\mathbf{b}\|_2^2$, we have $\nabla_{\mathbf{y}} \|\mathbf{S}\mathbf{U}\mathbf{y} - \mathbf{S}\mathbf{b}\|_2^2 = 2(\mathbf{S}\mathbf{U})^T (\mathbf{S}\mathbf{U}\tilde{\mathbf{y}}^* - \mathbf{S}\mathbf{b}) = \mathbf{0}$. Thus,

$$\|\mathbf{U}^T \mathbf{S}^T \mathbf{S} \mathbf{U}(\tilde{\mathbf{y}}^* - \mathbf{y}^*)\|_2^2 = \|\mathbf{U}^T \mathbf{S}^T (\mathbf{S}\mathbf{U}\tilde{\mathbf{y}}^* - \mathbf{S}\mathbf{b} + \mathbf{S}\mathbf{b} - \mathbf{S}\mathbf{U}\mathbf{y}^*)\|_2^2$$

$$= \|\mathbf{U}^T \mathbf{S}^T \mathbf{S}(\mathbf{b} - \mathbf{U}\mathbf{y}^*)\|_2^2.$$

Applying (B.3) with $\delta = 1/200$ and combining with (B.8) using union bound, we thus have that with probability $99/100$,

$$\|\tilde{\mathbf{y}}^* - \mathbf{y}^*\|_2^2 \le 4\|\mathbf{U}^T \mathbf{S}^T \mathbf{S} \mathbf{U}(\tilde{\mathbf{y}}^* - \mathbf{y}^*)\|_2^2 = 4\|\mathbf{U}^T \mathbf{S}^T \mathbf{S}(\mathbf{b} - \mathbf{U}\mathbf{y}^*)\|_2^2 \le 4\epsilon \|\mathbf{U}\mathbf{y}^* - \mathbf{b}\|_2^2. \tag{B.9}$$

Plugging into (B.7) and adjusting $\epsilon$ by a constant factor completes the proof of Theorem 1.1. $\quad\square$

### B.1 PROOF OF COROLLARY 1.1

We briefly comment on how to derive Corollary 1.1 from our main result. By definition of the method, we immediately have that our binary-tree-based pivotal sampling is $k$-homogeneous, so we just need to show that it produces a distribution over samples that is $\ell_\infty$-independent with constant parameter $D_{\text{inf}}$. This fact can be derived directly from a line of prior work. In particular, (Bränden and Jonasson, 2012) proves that binary-tree-based pivotal sampling satisfies negative association, (Pemantle and Peres, 2014) proves that negative association implies a stochastic covering property, and (Kaufman et al., 2022) shows that any distribution satisfying the stochastic covering property has $\ell_\infty$-independence parameter at most $D = 2$. We also give an arguably more direct alternative proof below based on a natural conditional variant of negative correlation.

*Proof of Corollary 1.1.* (Bränden and Jonasson, 2012) proves that binary-tree-based pivotal sampling is *conditionally negatively associated* (CNA). Given a set $\mathcal{C} \subseteq [n]$ and a vector $\mathbf{c} \in \{0,1\}^{|\mathcal{C}|}$, we denote the condition $\xi_c = c_i$ for all $i \in \mathcal{C}$ by $C$. Conditional negative association asserts that, for all $C$, any disjoint subsets $\mathcal{S}$ and $\mathcal{T}$ of $\{\xi_1, \cdots, \xi_n\}$, and any non-decreasing functions $f$ and $g$,

$$\mathbb{E}[f(\mathcal{S})|C] \cdot \mathbb{E}[g(\mathcal{T})|C] \geq \mathbb{E}[f(\mathcal{S})g(\mathcal{T})|C].$$

When $\mathcal{S}$ and $\mathcal{T}$ are singletons and $f$ and $g$ are the identity functions, we have

$$\mathbb{E}[\xi_i|C] \cdot \mathbb{E}[\xi_j|C] \geq \mathbb{E}[\xi_i\xi_j|C]. \tag{B.10}$$

Since $\mathbb{E}[\xi_i|C] = \Pr[\xi_i = 1|C]$, we also have

$$\Pr[\xi_i = 1|C]\Pr[\xi_j = 1|C] \geq \Pr[\xi_i = 1 \wedge \xi_j = 1|C]$$
$$\Pr[\xi_i = 1|C] \geq \frac{\Pr[\xi_i = 1 \wedge \xi_j = 1|C]}{\Pr[\xi_j = 1|C]} = \Pr[\xi_i = 1|\xi_j = 1 \wedge C]. \tag{B.11}$$

In words, the entries of our vector $\boldsymbol{\xi}$ are negatively correlated, even conditioned on fixing any subset of entries in the vector. We will use this fact to show that $\sum_{j \in [n]} |\mathcal{I}_\mu^{\mathcal{S}}(i,j)| \leq 2$ for all $i \in [n]$, where $\mathcal{I}_\mu^{\mathcal{S}}$ is as defined in Definition 3.1. For a fixed $i$, let $q_i = \Pr_{\boldsymbol{\xi} \sim \mu}[\xi_i = 1|\xi_\ell = 1 \forall \ell \in \mathcal{S}]$. Then, we have $|\mathcal{I}_\mu^{\mathcal{S}}(i,j)| = 0$ for $j \in \mathcal{S}$, $|\mathcal{I}_\mu^{\mathcal{S}}(i,j)| = 1 - q_i$ for $j = i$, and $\sum_{j \in [n] \backslash \mathcal{S} \cup \{i\}} |\mathcal{I}_\mu^{\mathcal{S}}(i,j)| = 1 - q_i$. The last fact follows from $k$-homogeneity, i.e. that $\sum_{i=1}^{n} q_i = k$, and (B.11), which implies that $\mathcal{I}_\mu^{\mathcal{S}}(i,j) \leq 0$ for all $j$ in $[n] \backslash \mathcal{S} \cup \{i\}$, so $\sum_{j \in [n] \backslash \mathcal{S} \cup \{i\}} |\mathcal{I}_\mu^{\mathcal{S}}(i,j)| = \left| \sum_{j \in [n] \backslash \mathcal{S} \cup \{i\}} \mathcal{I}_\mu^{\mathcal{S}}(i,j) \right|$. Thus, we have $\sum_{j \in [n]} |\mathcal{I}_\mu^{\mathcal{S}}(i,j)| = 2 - 2q_i \leq 2$. $\qquad\square$

## C   PROOF OF THEOREM 1.2

In this section we prove Theorem 1.2, which shows that pivotal obtains a better sample complexity for polynomial regression on an interval than independent leverage score sampling. Since we can always shift and scale our target function, without loss of generality we can take $[\ell, u]$ to be the interval $[-1, 1]$. We will sample from the infinite set of points on this interval. Each point $t \in [-1, 1]$ will correspond to a row in a regression matrix $\mathbf{A}$ with $d+1$ columns and an infinite number of rows. Such an object is sometimes referred to as a quasimatrix (Trefethen, 2009).[3] The row with index $t$ equals $\mathbf{a}_t = [1, t, t^2, \ldots, t^d]$. The leverage score for the row with index $t$ is defined analogously to Equation (1.2) as:

$$\tau(t) = \max_{\mathbf{x} \in \mathbb{R}^{d+1}} \frac{(\mathbf{x}^T \mathbf{a}_t)^2}{\int_{-1}^{1} (\mathbf{x}^T \mathbf{a}_s)^2 ds}. \tag{C.1}$$

Note that $\int_{-1}^{1} \tau(t)dt = (d+1)$ since $\mathbf{A}$ has $d + 1$ columns. We will sample $k$ points from $[-1, 1]$ interval with the probability of sampling point $t$ to be proportional to $\tau(t)$. To do so, we renormalize $\tau(t)$ and consider sampling point $t$ with probability proportional to $\frac{k\tau(t)}{d+1}$. This is the analog of sampling with probability $\tilde{p}_i$ in the discrete case, since $\int_{-1}^{1} \frac{k\tau(t)}{d+1} dt = k$. Then we apply pivotal

---

[3]We refer the reader to Avron et al. (2019), Erdélyi et al. (2020), or Chen and Price (2019) for a more in depth treatment of leverage score sampling and active linear regression for quasimatrices.

sampling in the infinite point limit, where we choose the pivotal competition order so that points in $[-1, 1]$ compete with each other from left to right across the interval. In this limit, the probability carried along in the competition will, at one point, accumulate to exactly 1. This defines the interval $I_1$ whose left endpoint is $-1$ and its right endpoint is defined so that $\int_{I_1} \frac{k\tau(t)}{d+1} dt = 1$. This means that the winner in the last competition in $I_1$ will be sampled. A new competition then starts for the next point, until the renormalized leverage score accumulates to 1 again and the winner is sampled. We can define $I_i (i = 1, 2, \ldots, k)$ to be adjacent intervals with $\int_{I_i} \frac{k\tau(t)}{d+1} dt = 1$. It can be seen that pivotal sampling in the infinite point limit will always sample *exactly one point* from each of $I_1, \ldots, I_k$. This is actually the only property of pivotal sampling we will need to prove Theorem 1.2, although we note that, within an interval $I_i$, point $t$ is selected with probability proportional to $\tau(t)/\int_{I_i} \tau(s)ds$.

To prove a sample complexity bound on $O(d/\epsilon)$ for polynomial regression, it suffices to prove that a constant factor subspace embedding guarantee holds when selecting $O(d)$ samples from $[-1, 1]$ using the pivotal sampling process above. Observe that the required $\epsilon$-error approximate matrix multiplication guarantee from Section 3 already holds with $O(d/\epsilon)$ samples collected via the pivotal method with leverage score marginals, and this analysis extends to the infinite quasi-matrix setting (see Appendix C.1). It is only the subspace embedding guarantee that adds an extra $\log d$ factor to the sample complexity. Since this factor is inherent to our use of a matrix Chernoff bound in analyzing the general case, to prove a tighter bound for polynomial regression we use a direct analysis that avoids matrix Chernoff entirely. In particular, our main result of this section is as follows:

**Theorem C.1.** *Let $k = O\left(\frac{d}{\alpha}\right)$ points $t_1, \ldots, t_k$ be selected from $[-1, 1]$ via pivotal sampling. Specifically, exactly one point $t_i$ is sampled from each interval $I_1, \ldots, I_k$ with probability proportional to its leverage score. We have (deterministically) that for any degree $d$ polynomial $p$, with $w(t) = \frac{\tau(t)}{d+1}$,*

$$\left| \int_{-1}^{1} p(t)^2 dt - \frac{1}{k} \sum_{i=1}^{k} \frac{p(t_i)^2}{w(t_i)} \right| \leq \alpha \int_{-1}^{1} p(t)^2 dt. \tag{C.2}$$

To translate from the notation above to the notation used in (B.1), note that $\int_{-1}^{1} p(t)^2 dt = \|\mathbf{U}\mathbf{x}\|_2^2$ where $\mathbf{U}$ is an orthogonal span for the quasi-matrix $\mathbf{A}$ defined above, and $\mathbf{x}$ is a $d+1$ dimensional vector containing coefficients of the polynomial $p$ in the basis $\mathbf{U}$. Correspondingly, $\frac{1}{k} \sum_{i=1}^{k} \frac{p(t_i)^2}{w(t_i)} = \|\mathbf{S}\mathbf{U}\mathbf{x}\|_2^2$.

*Proof of Theorem 1.2.* With Theorem C.1 providing the required subspace embedding guarantee from (B.1), the proof of Theorem 1.2 is essentially identical to the proof for the discrete case (Theorem 1.1). The only remaining requirement is the required approximate matrix-vector multiplication guarantee from (B.3). For completeness, we provide a direct proof in the polynomial regression case in Appendix C.1. □

We build up to the proof of Theorem C.1 by introducing several intermediate results. Our approach is inspired by a result of Kane et al. (2017), which also obtains $O(d/\epsilon)$ sample complexity for active degree-$d$ polynomial regression, albeit with a sampling distribution not based on leverage scores. In particular, they prove the same guarantees as Theorem C.1, but where points are selected uniformly from $k$ intervals that evenly partition the Chebyshev polynomial weight function $1/\sqrt{1-t^2}$. As we will see, the leverage scores for polynomials closely approximate this weight function. This connection is well known, as the leverage score function from (C.1) is exactly proportional to the inverse of the polynomial "Christoffel function" under the uniform measure, a well-studied function in approximation theory.

For simplicity, from now on we denote $f(t) = p(t)^2$ and describe the main structure of the proof of Theorem C.1. The left hand side of (C.2) can be decomposed into a sum of errors in individual intervals as

$$\left| \int_{-1}^{1} f(t)dt - \sum_{i=1}^{k} \frac{1}{k} \frac{f(t_i)}{w(t_i)} \right| \leq \sum_{i=1}^{k} \int_{I_i} \left| \frac{f(t)}{w(t)} - \frac{f(t_i)}{w(t_i)} \right| w(t)dt. \tag{C.3}$$

In each interval $I_i$, noting that by the definition of $I_i$, $\int_{I_i} w(t)dt = \frac{1}{k}$. So we have

$$
\int_{I_i} \left| \frac{f(t)}{w(t)} - \frac{f(t_i)}{w(t_i)} \right| w(t)dt = \int_{I_i} \left| \int_{t_i}^{t} \frac{f'(s)w(s) - f(s)w'(s)}{w^2(s)} ds \right| w(t)dt
$$
$$
\leq \int_{I_i} \left( \int_{I_i} \frac{|f'(s)|w(s) + f(s)|w'(s)|}{w^2(s)} ds \right) w(t)dt \qquad \text{(C.4)}
$$
$$
= \frac{1}{k} \int_{I_i} \frac{|f'(t)|}{w(t)} dt + \frac{1}{k} \int_{I_i} \frac{f(t)|w'(t)|}{w^2(t)} dt.
$$

Therefore,

$$
\left| \int_{-1}^{1} f(t)dt - \sum_{i=1}^{k} \frac{1}{k} \frac{f(t_i)}{w(t_i)} \right| \leq \frac{1}{k} \int_{-1}^{1} \frac{|f'(t)|}{w(t)} dt + \frac{1}{k} \int_{-1}^{1} \frac{f(t)|w'(t)|}{w^2(t)} dt. \qquad \text{(C.5)}
$$

Then, proving the upper bound in Theorem C.1 boils down to establishing a *lower bound* for $w(t)$, an *upper bound* for $|w'(t)|$, and the connection between $\int_{-1}^{1} \frac{|f'(t)|}{w(t)} dt$ and $\int_{-1}^{1} f(t)dt$, which is related to the lower bound for $w(t)$.

Both $f(t)$ and $w(t)$ are polynomials. This is key to providing both pointwise and (weighted) integral upper bounds of their derivatives by their function values and integrals. In polynomial approximation theory, Markov-Bernstein inequalities address exactly this point.

We begin with the integral form of the Markov-Bernstein inequalities from Borwein and Erdélyi (1995) and Nevai (1979).

**Proposition C.2** ($L^1$ Bernstein's inequality)**.** *For any degree $d$ polynomial $p(t)$, with a universal constant $C_0$,*

$$
\int_{-1}^{1} \left| \sqrt{1-t^2}\, p'(t) \right| dt \leq C_0\, d \int_{-1}^{1} |p(t)|dt. \qquad \text{(C.6)}
$$

**Proposition C.3** ($L^1$ Markov's inequality)**.** *For any degree $d$ polynomial $p(t)$, with a universal constant $C_1$,*

$$
\int_{-1}^{1} |p'(t)|\, dt \leq C_1\, d^2 \int_{-1}^{1} |p(t)|dt. \qquad \text{(C.7)}
$$

The integral on the left hand side of (C.6) is weighted by $\sqrt{1-t^2}$. This weight is negligible near the boundary of the interval $[-1, 1]$, which, to some extent, explains the milder $O(d)$ dependence on the right hand side of (C.6) compared to the $O(d^2)$ dependence in (C.7).

(C.6) and (C.7) also indicate that a lower bound on $w(t)$ of the form $\frac{1}{\sqrt{1-t^2}}$ or a constant can cooperate with $|f'(t)|$ to yield an integral bound. This is indeed realizable, and we will prove that the lower bound on $w(t)$ is of the two different forms in different parts of the interval $[-1, 1]$. It is roughly the minimum of these two, as $\frac{1}{\sqrt{1-t^2}}$ blows up near the boundary. We define the *middle region* to be a centered subinterval in $[-1, 1]$ that is $\sim \frac{1}{d^2}$ away from the boundary, and the *boundary region* to be the two subintervals of length $\sim \frac{1}{d^2}$ near the boundary. The exact expression of these two regions will be clear when we state different lower bounds of $w(t)$. There we note that these two regions are overlapping so that we get a lower bound on $w(t)$ on the whole interval. Matching this pattern of lower bounds, we will also apply different upper bounds of $w'(t)$ in these two regions.

**Lower bound on $w(t)$ in the middle region.** Erdélyi and Nevai (1992) states the following bound on $\tau(t)$, which shows the relation between the leverage score and the Chebyshev polynomial weight function. See Section 4.3 in Meyer et al. (2023) for further discussion of this result.

**Proposition C.4.** *With a constant $C_2 > 0$,*

$$
\tau(t) \geq \frac{C_2(d+1)}{\pi\sqrt{1-t^2}}, \quad w(t) \geq \frac{C_2}{\pi\sqrt{1-t^2}}, \quad \text{for } |t| \leq \sqrt{1 - \frac{9}{(d-1)^2}}. \qquad \text{(C.8)}
$$

**Lower bound on $w(t)$ in the boundary region.** This requires a bit more effort than the case of the middle region. Inspired by (C.7), we aim for the following bound on $\tau(t)$ and $w(t)$.

**Proposition C.5.** *With constants $C_3 > 0, c > \frac{9}{2}$,*

$$\tau(t) \geq C_3 d(d+1), \quad w(t) \geq C_3 d, \quad for \ |t| > 1 - \frac{c}{d^2}. \tag{C.9}$$

We specify $c > \frac{9}{2}$ because this implies that, at least for $d$ large enough, the middle region, defined as $\left[-\sqrt{1 - \frac{9}{(d-1)^2}}, \sqrt{1 - \frac{9}{(d-1)^2}}\right]$, and the boundary region, defined as $\left[-1, -1 + \frac{c}{d^2}\right] \cup \left[1 - \frac{c}{d^2}, 1\right]$, have overlap.

To prove Proposition C.5, we need to employ the explicit expression for $\tau(t)$ in terms of Legendre polynomials and apply properties of Legendre polynomials. We denote the unnormalized degree $d$ Legendre polynomial by $P_d(t)$ and we fix $P_d(1) = 1$. We use $L_d(t)$ to denote the normalized Legendre polynomials satisfying $\int_{-1}^{1} L_d^2(t) dt = 1$. There are classical, explicit bounds for $L_d(t)$ and its derivative on the interval $[-1, 1]$.

**Lemma C.6** (See e.g. Siegel (1955)). *For $|t| \leq 1$, the normalized degree $d$ Legendre polynomials $L_d$, satisfy*

$$|L_d(t)| \leq L_d(1) = \sqrt{d + \frac{1}{2}}, \quad |L_d'(t)| \leq \frac{d(d+1)}{2}\sqrt{d + \frac{1}{2}}. \tag{C.10}$$

*Proof of Proposition C.5:* Since $\tau(t)$ and $w(t)$ are related as $w(t) = \frac{\tau(t)}{d+1}$, we only need to prove the bound for $\tau(t)$. The proof follows from the fact that, $\tau(t)$ can be equivalently written as the squared norm of the $t$-th row in any orthonormal basis for $\mathbf{A}$,

$$\tau(t) = \sum_{i=0}^{d} L_i^2(t). \tag{C.11}$$

See e.g. Equation (4) in Meyer et al. (2023) for this equivalency. We will then lower bound $\tau(t)$ by deriving lower bounds for the individual summands, ultimately arguing that for $|t| > 1 - \frac{c}{d^2}$, $\tau(t) \gtrsim d^2$. Lower bounds on individual $L_i^2(t)$ can be derived with an upper bound on the derivative, since we know at the boundary $L_i(1) = \sqrt{i + \frac{1}{2}}$ exactly and $L_i^2(t)$ is an even function. In particular, by Lemma C.6,

$$|(L_i^2)'(t)| = |2L_i(t)L_i'(t)| \leq i(i + \frac{1}{2})(i+1).$$

Therefore, for any $c > 0$ and $|t| > 1 - \frac{c}{d^2}$,

$$L_i^2(t) \geq (i + \frac{1}{2}) - i(i + \frac{1}{2})(i+1)\frac{c}{d^2}.$$

For a given $c$, $L_i^2(t)$ is thus positive and of order $\gtrsim i$ as long as $i \leq d/c'$ for a sufficiently large constant $c' > 1$. A careful algebraic manipulation will show that we can choose $c > \frac{9}{2}$. With the classical formula for summing a linear growth sequence, we have for some constant $C_3$,

$$\tau(t) = \sum_{i=0}^{d} L_i^2(t) \geq \sum_{i=0}^{d/c'} L_i^2(t) \geq C_3 d(d+1). \qquad \square$$

Now we move on to upper bounds on $w'(t)$ in the two regions matching the form of $w(t)$ so that some cancellation can occur to leave a clean integral $\int_{I_i} f(t) dt$ in (C.5). The classical Markov-Bernstein inequalities in Borwein and Erdélyi (1995) bound the pointwise value of the derivative of a polynomial by its maximum function value.

**Lemma C.7** (Bernstein's inequality). *For any degree $d$ real polynomial $p(t)$,*

$$|p'(t)| \leq \frac{d}{\sqrt{1-t^2}} \sup_{t \in [-1,1]} |p(t)|, \quad for \ -1 < t < 1. \tag{C.12}$$

**Lemma C.8** (Markov's inequality). *For any degree $d$ real polynomial $p(t)$,*

$$|p'(t)| \leq d^2 \sup_{t \in [-1,1]} |p(t)|, \quad for \ -1 < t < 1. \tag{C.13}$$

Both Lemma C.7 and Lemma C.8 are valid on the whole interval $[-1, 1]$, but one of them is tighter than the other depending on where $t$ sits. Although $w'(t)$ itself is a degree $2d - 1$ polynomial, it turns out that the bounds directly given by Lemma C.7 and Lemma C.8 are too crude. However, we can unwrap more structure in $\tau'(t)$ by applying the recurrence relation of Legendre polynomials.

**Proposition C.9** (Neat expression of $\tau'(t)$)**.**

$$\tau'(t) = P'_{d+1}(t) P'_d(t). \tag{C.14}$$

*Proof.* We write $\tau(t)$ in terms of the unnormalized Legendre polynomials $P_n$ as

$$\tau(t) = \sum_{i=0}^{d} L_i^2(t) = \sum_{i=0}^{d} (i + \frac{1}{2}) P_i^2.$$

$P_n$ and $P'_n$ are related by the recurrence relation $(2n + 1)P_n = P'_{n+1} - P'_{n-1}$ with $P'_0 = 0$, so

$$\tau'(t) = \sum_{i=0}^{d} (2i + 1) P_i P'_i = \sum_{i=1}^{d} P'_i (P'_{i+1} - P'_{i-1}) = P'_{d+1} P'_d. \qquad \square$$

**Upper bound on $w'(t)$ in the middle region.**

**Proposition C.10.**

$$|\tau'(t)| \leq \frac{d(d+1)}{1 - t^2}, \quad |w'(t)| \leq \frac{d}{1 - t^2}, \quad for \ -1 < t < 1. \tag{C.15}$$

*Proof.* This follows from Proposition C.9 and Bernstein's inequality in Lemma C.7 for Legendre polynomials $P_d$ and $P_{d+1}$ by noting that the maximum of these polynomials are $P_i(1) = 1, \forall i$. $\quad \square$

**Upper bound on $w'(t)$ in the boundary region.** The bound in Proposition C.10 will blow up near the boundary. This is caused by the blowing up of Bernstein's inequality in Lemma C.7. We should use the tighter upper bound offered by Markov's inequality in Lemma C.8 and a similar proof will give the following bound.

**Proposition C.11.**

$$|\tau'(t)| \leq d^2(d+1)^2, \quad |w'(t)| \leq d^2(d+1), \quad for \ -1 \leq t \leq 1. \tag{C.16}$$

As mentioned, Proposition C.10 and Proposition C.11 cannot be obtained by treating $\tau(t)$ as a general $2d - 1$ polynomial. In fact, directly applying Lemma C.7 and Lemma C.8 to $\tau(t)$, we will get instead

$$|\tau'(t)| \leq \frac{(2d-1)(d+1)^2}{2\sqrt{1-t^2}} \quad \text{and} \quad |\tau'(t)| \leq \frac{1}{2}(2d-1)^2(d+1)^2,$$

which is significantly worse in the middle region.

With the previous intermediate results in place, we are now ready to prove our main claim, following the proof of Lemma 2.1 in Kane et al. (2017) for intervals defined by the Chebyshev measure.

*Proof of Theorem C.1.* Recall we start with (C.5) as

$$\left| \int_{-1}^{1} f(t) dt - \sum_{i=1}^{k} \frac{1}{k} \frac{f(t_i)}{w(t_i)} \right| \leq \frac{1}{k} \int_{-1}^{1} \frac{|f'(t)|}{w(t)} dt + \frac{1}{k} \int_{-1}^{1} \frac{f(t)|w'(t)|}{w^2(t)} dt. \tag{C.17}$$

We will use different pointwise bounds of $w(t)$ and $w'(t)$ in different parts of the interval. For the *middle region*, by Proposition C.4 and Proposition C.2 we have

$$\frac{1}{k}\int_{-\sqrt{1-\frac{9}{(d-1)^2}}}^{\sqrt{1-\frac{9}{(d-1)^2}}} \frac{|f'(t)|}{w(t)}dt \leq \frac{\pi}{C_2 k}\int_{-1}^{1}\sqrt{1-t^2}|f'(t)|dt \leq \frac{2\pi C_0 d}{C_2 k}\int_{-1}^{1}f(t)dt. \tag{C.18}$$

Also, by Proposition C.4 and Proposition C.10, $\frac{|w'(t)|}{w^2(t)} \leq \frac{\frac{d}{1-t^2}}{\frac{C_2^2}{\pi^2(1-t^2)}} = \frac{\pi^2 d}{C_2^2}$. Therefore,

$$\frac{1}{k}\int_{-\sqrt{1-\frac{9}{(d-1)^2}}}^{\sqrt{1-\frac{9}{(d-1)^2}}} \frac{f(t)|w'(t)|}{w^2(t)}dt \leq \frac{\pi^2 d}{C_2^2 k}\int_{-1}^{1}f(t)dt. \tag{C.19}$$

For the *boundary region*, by Proposition C.5 and Proposition C.3 we have

$$\frac{1}{k}\int_{[-1,-1+\frac{c}{d^2}]\cup[1-\frac{c}{d^2},1]} \frac{|f'(t)|}{w(t)}dt \leq \frac{1}{C_3 kd}\int_{-1}^{1}|f'(t)|dt \leq \frac{4C_1 d}{C_3 k}\int_{-1}^{1}|f(t)|dt. \tag{C.20}$$

Also, by Proposition C.5 and Proposition C.11, we have $\frac{|w'(t)|}{w^2(t)} \leq \frac{(d+1)d^2}{C_3^2 d^2} \leq \frac{2d}{C_3^2}$. Therefore,

$$\frac{1}{k}\int_{[-1,-1+\frac{c}{d^2}]\cup[1-\frac{c}{d^2},1]} \frac{f(t)|w'(t)|}{w^2(t)}dt \leq \frac{2d}{C_3^2 k}\int_{-1}^{1}f(t)dt. \tag{C.21}$$

Finally, noticing that the middle region $\left[-\sqrt{1-\frac{9}{(d-1)^2}}, \sqrt{1-\frac{9}{(d-1)^2}}\right]$ and the boundary region $\left[-1,-1+\frac{c}{d^2}\right] \cup \left[1-\frac{c}{d^2},1\right]$ are overlapping, we can further upper bound the right hand side of (C.17) by adding up (C.18), (C.19), (C.20), and (C.21), so that

$$\left|\int_{-1}^{1}f(t)dt - \sum_{i=1}^{k}\frac{1}{k}\frac{f(t_i)}{w(t_i)}\right| \leq \left(\frac{2\pi C_0}{C_2} + \frac{\pi^2}{C_2^2} + \frac{4C_1}{C_3} + \frac{2}{C_3^2}\right)\frac{d}{k}\int_{-1}^{1}f(t)dt. \tag{C.22}$$

Then, by taking $k = O(\frac{d}{\alpha})$, we prove the statement. $\qquad\square$

## C.1 DIRECT PROOF OF APPROXIMATE MATRIX-VECTOR MULTIPLICATION BOUND

*Approximate matrix-vector multiplication bound for Theorem 1.2.* Let $\mathbf{U}$ be an orthogonal basis of the column span of the polynomial regression quasi-matrix $\mathbf{A}$, as defined at the beginning of Appendix C. As in the finite dimensional setting, our goal is to prove that, if we take $k = O\left(\frac{d}{\epsilon\delta}\right)$ samples via pivotal sampling, using those samples to construct a sampling matrix $\mathbf{S}$, then with probability $1 - \delta$,

$$\|\mathbf{U}^T\mathbf{S}^T\mathbf{S}(\mathbf{b} - \mathbf{U}\mathbf{y}^*)\|_2^2 \leq \epsilon\|\mathbf{b} - \mathbf{U}\mathbf{y}^*\|_2^2. \tag{C.23}$$

We start with the key step (B.4) in the matrix case as

$$\Pr\left[\|\mathbf{U}^T\mathbf{S}^T\mathbf{S}(\mathbf{b} - \mathbf{U}\mathbf{y}^*)\|_2^2 \geq \epsilon\|\mathbf{b} - \mathbf{U}\mathbf{y}^*\|_2^2\right] \leq \frac{\mathbb{E}\left[\|\mathbf{U}^T\mathbf{S}^T\mathbf{S}(\mathbf{b} - \mathbf{U}\mathbf{y}^*)\|_2^2\right]}{\epsilon\|\mathbf{b} - \mathbf{U}\mathbf{y}^*\|_2^2}. \tag{C.24}$$

Our goal is to prove $\mathbb{E}\left[\|\mathbf{U}^T\mathbf{S}^T\mathbf{S}(\mathbf{b} - \mathbf{U}\mathbf{y}^*)\|_2^2\right] = O(\frac{d}{k})\|\mathbf{b} - \mathbf{U}\mathbf{y}^*\|_2^2$. We have that $\mathbf{U}$'s $i$-th column is the normalized Legendre polynomial $L_i$. Let $\mathbf{p}^* = \arg\min_{\text{degree } d \text{ polynomial } \mathbf{p}} \|\mathbf{p} - \mathbf{b}\|_2^2$. Then $\mathbf{U}\mathbf{y}^* = \mathbf{p}^*$, where $\mathbf{y}^*$ is a length $d+1$ vector holding the coefficients of $\mathbf{p}^*$ when expanded in the basis of Legendre polynomial $L_i$'s. So the expectation can be written as

$$\mathbb{E}\left[\|\mathbf{U}^T\mathbf{S}^T\mathbf{S}(\mathbf{b} - \mathbf{U}\mathbf{y}^*)\|_2^2\right] = \mathbb{E}\left[\sum_{j=0}^{d}\left(\sum_{i=1}^{k}\frac{L_j(t_i)(p^*(t_i) - b(t_i))}{kw(t_i)}\right)^2\right]. \tag{C.25}$$

Note that each $t_i$ is chosen independently from $I_i$ with probability density function in $I_i$ as $\tau(t)/\int_{I_i} \tau(t)dt = kw(t)$. So we can calculate the expectation in (C.25) to get

$$
\begin{aligned}
\mathbb{E}\big[\|\mathbf{U}^T\mathbf{S}^T\mathbf{S}(\mathbf{b} - \mathbf{U}\mathbf{y}^*)\|_2^2\big] &= \mathbb{E}\left[\sum_{j=0}^{d}\sum_{i=1}^{k}\left(\frac{L_j(t_i)(p^*(t_i) - b(t_i))}{kw(t_i)}\right)^2\right] \\
&= \sum_{j=0}^{d}\sum_{i=1}^{k}\int_{I_i}\frac{(L_j(t)(p^*(t) - b(t)))^2}{kw(t)}dt \\
&= \sum_{i=1}^{d}\int_{I_i}\frac{\tau(t)(p^*(t) - b(t))^2}{kw(t)}dt \\
&= \frac{d+1}{k}\int_{-1}^{1}(p^*(t) - b(t))^2 dt,
\end{aligned}
\tag{C.26}
$$

where we use (C.11) in the third equality. The integral in the last line is exactly $\|\mathbf{b} - \mathbf{U}\mathbf{y}^*\|_2^2$, so we prove that $\mathbb{E}\big[\|\mathbf{U}^T\mathbf{S}^T\mathbf{S}(\mathbf{b} - \mathbf{U}\mathbf{y}^*)\|_2^2\big] = \frac{d+1}{k}\|\mathbf{b} - \mathbf{U}\mathbf{y}^*\|_2^2$. Plugging this back into (C.24), we get

$$
\Pr\big[\|\mathbf{U}^T\mathbf{S}^T\mathbf{S}(\mathbf{b} - \mathbf{U}\mathbf{y}^*)\|_2^2 \geq \epsilon\|\mathbf{b} - \mathbf{U}\mathbf{y}^*\|_2^2\big] \leq \frac{d+1}{\epsilon k}.
\tag{C.27}
$$

This shows that if we take $k = O(\frac{d}{\epsilon\delta})$ samples via pivotal sampling, then with probability $1 - \delta$, $\|\mathbf{U}^T\mathbf{S}^T\mathbf{S}(\mathbf{b} - \mathbf{U}\mathbf{y}^*)\|_2^2 \leq \epsilon\|\mathbf{b} - \mathbf{U}\mathbf{y}^*\|_2^2$. $\qquad\square$

## D    COMPLEMENTARY EXPERIMENTS

In Section 4, we conduct experiments using three targets; a damped harmonic oscillator, a heat equation, and a chemical surface reaction. Here, we show the results of additional simulations. Thus far, the original domain of the experiments is 2D for visualization purposes. Here, we consider the 3D original domain by freeing one more parameter in the damped harmonic oscillator model. We also summarize the number of samples required to achieve a given target error for all four test problems. To corroborate the discussion in Section 1.1, we provide a simulation result showing that leverage score sampling outperforms uniform sampling. Lastly, the performance of the randomized BSS algorithm from Chen and Price (2019) on the 2D damped harmonic oscillator target is displayed which supports our discussion in Section 1.3. Before showing the simulation results, we begin this section with a deferred detail of the chemical surface coverage target in Section 4.

**Chemical Surface Coverage.** As explained in (Hampton and Doostan, 2015), the target function models the surface coverage of certain chemical species and considers the uncertainty $\rho$ which is parameterized by absorption $\alpha$, desorption $\gamma$, the reaction rate constant $\kappa$, and time $t$. Given a position $(x, y)$ in 2D space, our quantity of interest $\rho$ is modeled by the non-linear evolution equation:

$$
\begin{aligned}
\frac{d\rho}{dt} &= \alpha(1 - \rho) - \gamma\rho - \kappa(1 - \rho)^2\rho, \quad \rho(t = 0) = 0.9, \\
\alpha &= 0.1 + \exp(0.05x), \quad \gamma = 0.001 + 0.01\exp(0.05y)
\end{aligned}
\tag{D.1}
$$

In this experiment, we set $\kappa = 10$ and focus on $\rho$ after $t = 4$ seconds.

**3D Damped Harmonic Oscillator.** The damped harmonic oscillator model is given as (restated),

$$
\frac{d^2 x}{dt^2}(t) + c\frac{dx}{dt}(t) + kx(t) = f\cos(\omega t), \quad x(0) = x_0, \quad \frac{dx}{dt}(0) = x_1
\tag{D.2}
$$

In Section 4, we define the domain as $k \times \omega = [1, 3] \times [0, 2]$ while $f$ was fixed as $f = 0.5$. This time, we extend it to 3D space by setting the domain as $k \times f \times \omega = [1, 3] \times [0, 2] \times [0, 2]$. 3D plot of the target is given in Figure 7 (a). Note that if we slice the cube at $f' = 0.5$, we obtain the target in Section 4. This time, we create the base data matrix $\mathbf{A}' \in \mathbb{R}^{n \times d'}$ by constructing a $n = 51^3$ fine grid, and the polynomial degree is set to 12. Figure 7 (b) and (c) give examples of the leverage score of this data matrix.

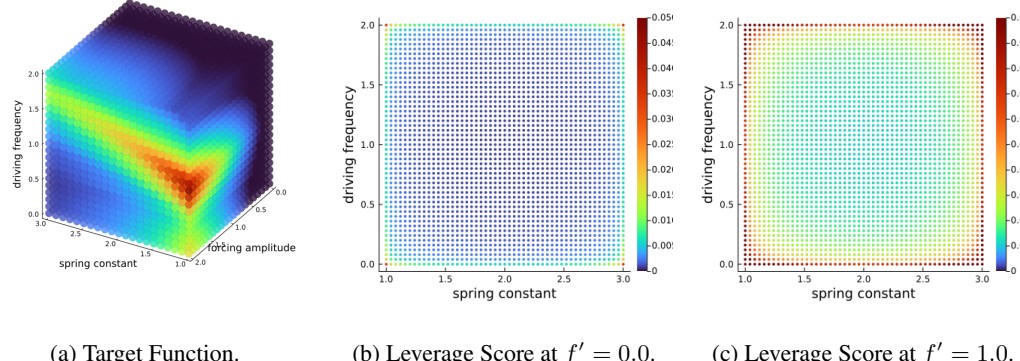

(a) Target Function.    (b) Leverage Score at $f' = 0.0$.    (c) Leverage Score at $f' = 1.0$.

Figure 7: (a): The target value of the 3D damped harmonic oscillator model. (b) and (c): Its leverage score. We have $f \in [0, 2]$, and the grid is sliced at $f = 0.5$ in (b) and at $f = 1.0$ in (c).

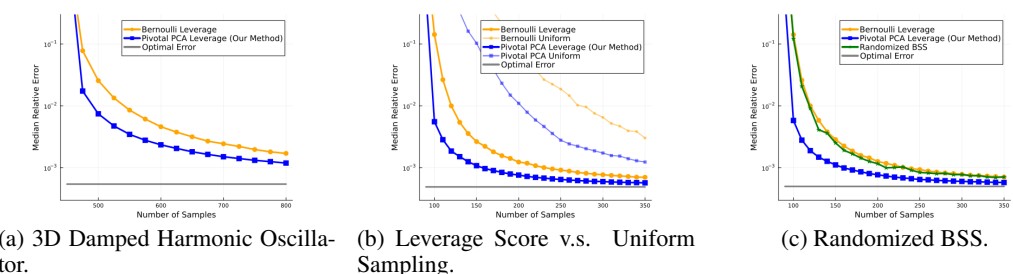

(a) 3D Damped Harmonic Oscillator.

(b) Leverage Score v.s. Uniform Sampling.

(c) Randomized BSS.

Figure 8: (a): Results for degree 12 active polynomial regression for the damped harmonic oscillator QoI in 3D space. (b), (c): Degree 12 active polynomial regression for the damped harmonic oscillator QoI in 2D space. (b) includes the performance of the Bernoulli sampling and our pivotal sampling both using uniform inclusion probability which is given with the thin lines. (c) demonstrates the approximation power of the randomized BSS algorithm. We run the sampling 2000 times with different parameters, round the number of samples to the tens place, and take the median error.

Figure 8 (a) shows the relative error of Bernoulli sampling and our pivotal sampling both using the leverage score. As expected, our method shows a better fit than Bernoulli sampling again.

**Samples Needed for a Certain Error.** Table 1 and 2 summarize the number of samples required to achieve $2 \times \mathrm{OPT}$ error and $1.1 \times \mathrm{OPT}$ error where OPT is the error we could obtain when all the data are labeled. In all four test problems, pivotal sampling achieves the target error with fewer samples than the existing method (independent leverlage scores sampling), which is denoted by Bernoulli in the tables. Our method is especially efficient when we aim at the target error close to OPT. For instance, to achieve the $1.1 \times \mathrm{OPT}$ error in the 2D damped harmonic oscillator model, our method requires less than half samples that Bernoulli sampling requires, showing a significant reduction in terms of the number of samples needed.

**Leverage Score Sampling vs. Uniform Sampling.** We also conduct a complementary experiment to show empirically that leverage score sampling is much more powerful than uniform sampling. We extend the simulation in Figure 1 to a uniform inclusion probability setting. This time, we draw 350 samples, repeat the simulation 100 times, and report the approximation with a median error in Figure 9. The result shows poor performance of the uniform sampling. As they draw fewer samples near the boundaries compared to leverage score sampling, they are not able to pin down the

|  | Oscillator 2D | Heat Eq. | Surface Reaction | Oscillator 3D |
|---|---|---|---|---|
| n | 10000 | 10000 | 10000 | $51^3$ |
| poly. deg. | 20 | 20 | 20 | 10 |
| Bernoulli (a) | 574 | 554 | 545 | 671 |
| Pivotal (b) | 398 | 395 | 390 | 533 |
| Efficiency (b / a) | 0.693 | 0.713 | 0.716 | 0.794 |

Table 1: Number of samples needed to achieve $2 \times \mathrm{OPT}$ error.

|  | Oscillator 2D | Heat Eq. | Surface Reaction | Oscillator 3D |
|---|---|---|---|---|
| n | 10000 | 10000 | 10000 | $51^3$ |
| poly. deg. | 12 | 12 | 12 | 10 |
| Bernoulli (a) | 924 | 814 | 903 | 3121 |
| Pivotal (b) | 450 | 442 | 492 | 1943 |
| Efficiency (b / a) | 0.487 | 0.523 | 0.545 | 0.623 |

Table 2: Number of samples needed to achieve $1.1 \times$ OPT error.

polynomial function near the edges, resulting in suffering large errors in these areas. The relative error plot is given in Figure 8 (b). We point to three observations. 1) Comparing the thick lines and the thin lines, one can tell that the use of leverage score significantly improves performance. 2) Comparing the orange lines and the blue lines, one can see that our *spatially-aware* pivotal sampling outperforms the Bernoulli sampling. 3) By combining the leverage score and *spatially aware* pivotal sampling (our method), one can attain the best approximation among the four sampling strategies.



(a) Bernoulli Uniform. (b) Bernoulli Leverage. (c) Pivotal Uniform. (d) Pivotal Leverage.

Figure 9: Visualizations of a polynomial approximation to the maximum displacement of a damped harmonic oscillator, as a function of driving frequency and spring constant. $\mathcal{X}$ is a uniform distribution over a box. All four draw 350 samples but by different sampling methods; (a) and (b) use Bernoulli sampling but (c) and (d) use pivotal sampling. Also, (a) and (c) select samples with uniform probability while (b) and (d) employs the leverage score. Clearly, the leverage score successfully pins down the polynomial function near the boundaries, resulting in better approximation.

**Randomized BSS Algorithm.** Finally, we run the randomized BSS algorithm (Chen and Price, 2019) on the 2D damped harmonic oscillator target function, and report the relative error together with Bernoulli leverage score sampling and our pivotal sampling in Figure 8 (c). The setting is the same as the Section 4 that the initial data points are drawn uniformly at random from $[1, 3] \times [0, 2]$, and the polynomial degree is set to 12. Even though this algorithm also has a theoretical sample complexity of $O(d/\epsilon)$ for the regression guarantee in (1.3), our sampling method achieves a certain relative error with fewer samples.

