# OpenReview forum: "Improved Active Learning via Dependent Leverage Score Sampling"
_ICLR.cc/2024/Conference — ICLR 2024 oral_

### Official Review · Reviewer_gm2B · 2023-10-17

**Soundness:** 3 good
**Presentation:** 3 good
**Contribution:** 3 good
**Rating:** 8
**Confidence:** 4

**Summary:**

This paper proposes a novel leverage score sampling method based on spatial pivoting and therefore the samples are dependent. The main results involve a nearly optimal sample complexity of $O(d\log d+d/\epsilon)$ for active regression which matches the standard independent leverage score sampling, and a optimal bound of $O(d/\epsilon)$ for polynomial regression. The key proof ingredient for the first result is an adaptation of the $\ell_\infty$ independence result due to [KKS22] and how to generalize this property to approximate matrix product, and for the second result is similar to [KKP17]. They also perform extensive experiments for active regressions on PDEs and the experiments are convincing enough that their algorithm has good performance.

**Strengths:**

While the techniques are not completely novel, I do like the theoretical implications of this paper --- they essentially show that leverage score sampling with limited independence also provides theoretically matching bound compared to independent leverage score sampling. This inspires one to design leverage score sampling algorithm that can adaptively choose the samples based on particular structure of the problem. They also show a BSS-type result for polynomial regression without resorting to the BSS barrier function argument. This is very interesting because as far as I know, all sparsification results with $O(d/\epsilon)$ sample complexity are more or less variants of BSS. I believe results and techniques in this paper have further applications in other sparsification problems.

In addition, they perform many numerical experiments to show that their algorithm works well in practice. This is nice as it aligns with the idea that one can modify leverage score sampling based on problems to get better (practical) algorithms.

**Weaknesses:**

As noted in the strengths part, the techniques in this paper are not very novel. To get their first $d\log d+d/\epsilon$ sample complexity for active regression, they mainly utilize [KKS22]. The approximate matrix product result requires a bit more work, but it's also standard and not surprising. For the second result regarding polynomial regression, it also mainly follows procedures from [KKP17]. Nevertheless, I still think these results are nice, and are interesting enough to be published in conferences like ICLR.

**Questions:**

A few comments regarding citation: the paper [KKS22] was published in SODA 2022. It's better to cite the proceeding version.

A question: it seems the binary tree-based approach to pre-partition the space based on the structure of the problem can also be extended to BSS-type sampling, e.g., see the data structures and algorithms in [SXZ22]. Do you think your argument and techniques can be generalized to BSS, to achieve an optimal sample complexity while leveraging particular structures?

[SXZ22]: Z. Song, Z. Xu and L. Zhang. Speeding up sparsification using inner product search data structures, 2022.

---

> ### Author Response · Authors · 2023-11-18
>
> Thank you very much for the positive review of our work. We will fix the issue with the citation, and also take a closer look at the SXZ22 paper you reference. We think it would be very exciting to come up with a method that matches the O(d/epsilon) theoretical guarantee of Randomized BSS but also performs better than i.i.d. leverage score sampling in practice. We are honestly unsure if our pivotal sampling strategy could lead to such a method, as our improved bound for polynomial regression very strongly relies on additional structure in that problem.

---

### Official Review · Reviewer_gr3D · 2023-10-31

**Soundness:** 3 good
**Presentation:** 4 excellent
**Contribution:** 3 good
**Rating:** 8
**Confidence:** 4

**Summary:**

This paper proposes a way to perform leverage score sampling that promotes spatial coverage for the problem of active linear regression. Motivated by the empirical successes of deterministic grid-based approaches to PDEs, the presented approach retains the strong theoretical guarantees of leverage score sampling with the spatially-well distributed samples of grid-based approaches. The key insight is to use a pivotal sampling algorithm where a binary tree that matches the geometry of the data is constructed deterministically and the samples are percolated up via head-to-head comparisons (probability weighted coin flip) at sibling nodes. This leads to the desirable behavior of spatial coverage since close neighbors in the tree are less likely to be both included. The authors conduct empirical evaluations that show the improved effectiveness of the proposed method.

**Strengths:**

* The problem of active linear regression is interesting and has applications to uncertainty quantification and parametric partial differential equations.
* The paper is very well-written and organized. There is a great overview of prior work as well as the contextualization of the results (e.g., the fact that the work builds on top of the recent result in Chernoff bounds under $\ell_\infty$ independence).
* The claims of the paper are adequately supported with rigorous theoretical analysis and empirical evaluations.
* The algorithm and its analysis is novel to the best of my knowledge. The presented analysis is general enough that it could be applied to any non-independent leverage score sampling method that obeys a weak one-sided  $\ell_\infty$ independence condition. This might be of independent interest to other researchers in the area. Leveraging
* The empirical evaluations are compelling and clearly show the improved effectiveness of the dependent leverage score sampling method.

**Weaknesses:**

* The presented analysis does not provide justification for the improved empirical effectiveness of the method (besides for $\ell_2$ polynomial regression). So, it is not clear theoretically why the method performs much better than independent leverage score sampling. Nevertheless, this is a limitation that the authors clearly concede and mention a few times throughout.

**Questions:**

The authors note that the algorithm of Chen and Price, 2019 is provably optimal, but that “in our initial experiments, it was not competitive with leverage score sampling in practice.” Why were these results not included in the main body of the paper? My understanding is that the Randomized BSS algorithm is only included as a result in Figure 8c of the appendix. It would be compelling to have comparisons to Chen and Price’s method in Fig. 4.

---

> ### Author Response · Authors · 2023-11-18
>
> Thank you for the supportive review and positive feedback.
>
> A few reviewers asked about comparison to the Randomized BSS method from [Chen, Price, 2019] and we will consider elevating the experiment from the appendix and adding additional results. One thing we want to note is that Randomized BSS is quite complex, and involves several parameters that must be tuned well for the method to achieve low sample complexity. For the experiment in the appendix, we tuned these parameters optimally for one specific problem, and the method was able to slightly outperform leverage score sampling. However, our best general purpose implementation (with parameters fixed across different problems) performed worse than the baseline i.i.d. leverage score sampling, which is why we did not include the results to begin with. We are not aware of other implementations or experimental evaluations of the method. An exciting direction for future work is to find a method that matches the tighter theoretical guarantees of Randomized BSS (for general regression problems) while also outperforming i.i.d. leverage score sampling in practice.

---

### Official Review · Reviewer_BstN · 2023-11-01

**Soundness:** 3 good
**Presentation:** 3 good
**Contribution:** 3 good
**Rating:** 6
**Confidence:** 3

**Summary:**

This paper investigates the active regression problem in the adversarial setting. The authors introduce a novel sampling methodology that promotes spatial coverage, combining it with leverage score sampling. The paper offers two theoretical contributions: the first establishes a sample complexity bound under the one-sided $\ell_{\infty}$ independence condition, while the second presents a bound applicable to a specific case of polynomial regression. Empirical results further validate the effectiveness of the proposed approach.

**Strengths:**

a. The proposed pivotal sampling method using a binary tree tournament is interesting and innovative.

b. The theoretical results are solid. The sample complexity bound for polynomial regression showcases an improvement of a logarithmic factor over the general case. The techniques employed in the analysis have potential implications beyond this specific research.

c. The authors provide empirical results to validate the effectiveness of the proposed pivotal sampling. These results clearly indicate that pivotal sampling significantly outperforms Bernoulli leverage sampling.

d. The presentation is clear and easy to follow.

**Weaknesses:**

a. My primary concern pertains to the computational complexity of the proposed method. The paper lacks both theoretical analysis and empirical results in this regard, leaving an important aspect unaddressed.

b. The experimental setup in the paper includes only a single baseline, and it would be beneficial to include additional baseline methods, such as maximum leverage score sampling.

**Questions:**

a. Is there any theoretical analysis or empirical results regarding the computational complexity of pivotal sampling, particularly concerning the construction of the binary tree?

b. How does the proposed pivotal sampling method compare with maximum leverage sampling strategy?

---

> ### Author Response · Authors · 2023-11-18
>
> Thank you for the supportive review and recognizing the methodological and theoretical contributions of the work. Reviewer hDW2 also asked about runtime, so I will paste the response below. We agree that the paper would benefit from increased discussion and experimental results on runtime.
>
> We will also consider additional baselines. We did implement the randomized BSS method from [Chen, Price 2019], although found it was uncompetitive (there is an experiment in Appendix D). Another option would be to consider e.g. square root leverage scores sampling, which was discussed in “A Statistical Perspective on Algorithmic Leveraging” by Ma, Mahoney, Yu. We are not 100% sure what method the reviewer is referring to as “maximum leverage score sampling”. Do you mean the method from the paper “Provable Deterministic Leverage Score Sampling” by Papailiopoulos, Kyrillidis, and Boutsidis? That paper selects the rows in A with the *largest* leverage scores. While this method works well in many settings, it performs very poorly for polynomial regression and related “continuous” problems where nearby points typically have similar leverage scores. For example, for polynomial regression, maximum leverage score sampling would *only* select samples right near the edge of the interval or box that we are fitting the function over.
>
> Comments on runtime to Reviewer hDW2:
> On average, we found that our PCA-based pivotal methods run in approximately 125% of the time of the baseline i.i.d. leverage score sampling method and the coordinate-wise splitting method runs in 114% of the time. So, the PCA method is slower, but the overhead is low overall.
>
> The reason for the low-overhead is that PCA is being run on *low-dimensional* data points. E.g., for a degree s polynomials in q=2 dimensions, PCA is being run on q=2 dimensional points, even though the dimension of the polynomial regression problem is d = O(s^2) (in our experiments, d = 60+). Additionally, since the size of the PCA problems is decreasing by a factor of 2 at every step of the splitting method, the total runtime cost of constructing the pivotal sampling tree is roughly O(nq^2log n). Accordingly, we found that the complexity of our method was dominated by the cost of computing leverage scores for the matrix A, which naively takes O(nd^2). This cost is also incurred by naive i.i.d. sampling. We do note that faster algorithms for approximating leverage scores are well studied and can be used instead. See e.g., the paper “Fast approximation of matrix coherence and statistical leverage” by Drineas, Magdon-Ismail, Mahoney, and Woodruff, and follow-up work, which improves the cost to time O(nd + d^3).

---

> > ### Comment · Reviewer_BstN · 2023-11-23
> >
> > Thanks for your response. I will keep my score.

---

### Official Review · Reviewer_hDW2 · 2023-11-01

**Soundness:** 3 good
**Presentation:** 4 excellent
**Contribution:** 3 good
**Rating:** 6
**Confidence:** 4

**Summary:**

In this work, the authors improve on active learning methods for linear and polynomial regressions under an agnostic noise setting. Here, the goal is to learn $x \in \mathbb{R}^d$ such that $Ax ≈ b$ while observing only a few entries of $b$. Prior works on active linear regression, studied for decades, assume $b$ to be equal to $Ax^*$ plus i.i.d. random noise. In this case, the problem can be addressed using tools from optimal experimental design. In the agnostic case, where such a noise model is not specified, near-optimal sample complexity results and use statistical leverage score-based sampling. In this work, the authors argue that "any" sampling strategy that has marginals proportional to leverage scores, satisfying a weak independence condition, will only need samples equal to those required using independent leverage score-based sampling up to constants (theoretical), given by $O(d\log d)$. They then propose a pivotal sampling-based approach (satisfying the criteria listed) which empirically performs much better, and also show theoretical improvements for the special case of polynomial regression, and uses $O(d)$ samples.

**Strengths:**

— The paper is extremely well-written, providing a thorough discussion on the advantages and limitations of the authors’ work alongside prior research.

— The result on "any" sampling strategy is very interesting, even if it doesn’t improve upon previous theoretical guarantees for the given scenario.

— Most sampling-based results obscure significant constants, as noted by the authors regarding the theoretically optimal result due to [Chen and Price 2019], which, in practice, performs poorly. Therefore, the authors' provision of approaches with improved empirical performance, even with similar theoretical guarantees is an important contribution.

**Weaknesses:**

— The empirical evaluation lacks clarity regarding the improvements. The authors should explicitly state the claimed 50% reduction in samples in the technical sections.

— The theoretical improvements are solely for polynomial regression, which is essentially linear regression with an infinite number of rows. It would be interesting to explore whether the pivotal sampling method has other implications for different norms. For instance, can this "tournament" style technique be applied to all norms and their respective optimal sampling strategies?

— The selection of the tree in Algorithm 1 significantly impacts the performance of the sampling method. If the running time is a major concern in large-scale systems or datasets, it isn’t clear if a PCA-based approach is always feasible.

**Questions:**

Pg 3. Theoretic -> theoretical

Pg 6. Bernouilli -> Bernoulli

As constants play a big role in the motivation of this work and their empirical improvements, a suggestion would be to note them down explicitly in the current work if possible.

The $\ell_\infty$ based independence condition. Can the main theorem result be stated in terms of the parameter $D_{inf}$? It would also be interesting to note the value in the case of independent uniform sampling. It suggests that the $D_{inf}$ is the same for any independent sampling approach.

In the experiments, it would be interesting to compare with [Chen and Price 2019] and also add a random permutation of rows as the leaves of the tree, in addition to PCA-based ones.

---

> ### Author Response · Authors · 2023-11-18
>
> Thank you for the accurate summary of our contributions and suggestions. We respond to specific comments below, but please let us know if you have any additional questions.
>
> – The empirical evaluation lacks clarity regarding the improvements…
> We agree with the suggestion here. The 50% improvement can be seen most clearly in Appendix D, where Table 1 and Table 2 compare the number of samples needed to achieve a given level of error. We are tight on space, but will plan on moving at least one of these tables up to the main experiments section in the final version of the paper.
>
> – It would be interesting to explore whether the pivotal sampling method has other implications for different norms.
> We think this is a really nice question for future work, especially given recent progress on understanding optimal sampling results for polynomial regression in various norms (see e.g. the cited work by [Meyer et al. 2023]). We would optimistically predict that we could similarly remove log(d) factors using tournament style methods like pivotal sampling, although the theoretical analysis will have to differ significantly. It would also be interesting to establish a general result like Theorem 1.1 for other norms, e.g. for lp norms, for which the optimal sampling probabilities are based on lp Lewis weights.
>
> – If the running time is a major concern in large-scale systems or datasets, it isn’t clear if a PCA-based approach is always feasible.
> We will plan on adding some discussion on runtime to the final version of the paper. We can also add some basic experimental results. On average, we found that our PCA-based pivotal methods run in approximately 125% of the time of the baseline i.i.d. leverage score sampling method and the coordinate-wise splitting method runs in 114% of the time. So, the PCA method is slower as the reviewer notes, but the overhead is low overall.
>
> The reason for the low-overhead is that PCA is being run on *low-dimensional* data points. E.g., for a degree s polynomials in q=2 dimensions, PCA is being run on q=2 dimensional points, even though the dimension of the polynomial regression problem is d = O(s^2) (in our experiments, d = 60+). Additionally, since the size of the PCA problems is decreasing by a factor of 2 at every step of the splitting method, the total runtime cost of constructing the pivotal sampling tree is roughly O(nq^2log n). Accordingly, we found that the complexity of our method was dominated by the cost of computing leverage scores for the matrix A, which naively takes O(nd^2). This cost is also incurred by naive i.i.d. sampling. We do note that faster algorithms for approximating leverage scores are well studied and can be used instead. See e.g., the paper “Fast approximation of matrix coherence and statistical leverage” by Drineas, Magdon-Ismail, Mahoney, and Woodruff, and follow-up work, which improves the cost to time O(nd + d^3).
>
> - Can the main theorem be stated in terms of the parameter D_inf?
> This is a good suggestion, and we will plan on doing so, as well as adding explicit constants as the reviewer suggestions. The final sample complexity will be c*(d log d)*(1/D_inf^2) + (d/epsilon)*(1/D_inf).
>
> Finally, we thank the reviewer for suggestions on experiments to add, and will consider the random permutation method as a baseline. We actually did implement the [Chen, Price 2019] method, and include one experiment in the appendix (Figure 8,c). We spent quite a bit of time optimizing the choice of constants in that method. We were ultimately able to beat i.i.d leverage score sampling, but only by a little bit. The method requires notably more samples than pivotal sampling.

---

> > ### Comment · Reviewer_hDW2 · 2023-11-30
> > **Reviewer response**
> >
> > I want to thank the authors for answering my questions -- they were really helpful. I plan to continue supporting the work with the same score.

---

### Official Review · Reviewer_Q8k3 · 2023-11-08

**Soundness:** 4 excellent
**Presentation:** 3 good
**Contribution:** 3 good
**Rating:** 8
**Confidence:** 2

**Summary:**

The authors suggest a modification of the pivotal sampling algorithm which allows for better domain coverage in such an applications as approximating PDE solutions.

**Strengths:**

The paper is well-written with clear motivation for the main theoretical results and nice experimental illustrations. The modified pivotal sampling algorithm (Algorithm 2) is appealing and seems to be quite easy to implement.

**Weaknesses:**

My main concerns are stated below and are related with the technical novelty of the results of the analysis of Theorems~1.1 and 1.2. The experimental section is sufficient for the demonstration of the approach, yet it could be elaborated to the higher-dimensional problems.

**Questions:**

I would like the authors to clarify the novelty of the suggested theoretical analysis. In particular, I do not understand why the auxiliary results for the proof of Theorem 1.1 do not follows directly from the literature? In particular, why the result of Lemma B.1 does not follow from the existing matrix Bernstein inequalities, see e.g. [Vershynin, Section 5.4]. The results on the approximate matrix multiplication (see the same appendix) also seems to be known [Tropp, 2012, Section 6.4]. It would be great if the authors could better elaborate on the novely, and technical novelty, of Theorems~1.1 and 1.2.

References:
[Vershynin, 2018] Vershynin, Roman. High-dimensional probability: An introduction with applications in data science. Vol. 47. Cambridge university press, 2018. https://www.math.uci.edu/~rvershyn/papers/HDP-book/HDP-book.pdf
[Tropp, 2015] Tropp, Joel A. "An introduction to matrix concentration inequalities." Foundations and Trends® in Machine Learning 8.1-2 (2015): 1-230. https://arxiv.org/pdf/1501.01571.pdf

---

> ### Author Response · Authors · 2023-11-18
>
> Thank you for the thoughtful response. We respond below to your questions about the connection of our results to prior work. Please let us know if you have any additional questions.
>
> The first Lemma asked about, Lemma B.1, is not our result, but rather a result we use from [Kaufman et al., 2022]. This lemma is a strong generalization of prior matrix Bernstein inequalities (including the one cited by the reviewer) because prior matrix Bernstein bounds only apply when Y_1, …, Y_n are sampled *independently*. Lemma B.1 applies to *non-independent* sampling distributions (like pivotal sampling) as long as the distribution is homogenous and satisfies the l_infinity independence guarantees.  [Kaufman et al., 2022]’s result is the culmination of a line of work on relaxing the independence requirement. For example, prior work in [Kyng, Song, 2018] proved matrix concentration inequalities for non-independent strongly Rayleigh distributions, which is a subset of l_infinity independent distributions. Similarly, approximate matrix-multiplication results were known for independent sampling, but not for dependent distributions. Our main technical contribution in proving Theorem 1.1 is to show that approximate matrix-multiplication can *also* be generalized to the case when sampling is dependent, but satisfies l_infinity independence. This proof is on Page 14 and 15 of the appendix. Combining with the existing results from [Kaufman et al., 2022] establishes Theorem 1.1. Finally, we stress that some sort of correlation (non-independence) between samples is necessary to produce a spatially well-balanced distribution.
>
> The contribution of Theorem 1.2 differs from that of Theorem 1.1 because the result of Theorem 1.2 (which is specialized to fitting polynomials) has sample complexity O(d) instead of O(dlogd). Independent leverage score sampling requires O(dlogd) samples for fitting degree d polynomials, so Theorem 1.2 shows a concrete setting where pivotal sampling gives a provably stronger bound (i.e., it eliminates a log d). To prove this bound, we cannot use matrix concentration inequalities at all (even those of [Kaufman et al., 2022])  since these bounds inherently involve a log d factor. So, we instead prove the required subspace embedding guarantee “from scratch” using tools from polynomial approximation theory. For example, one ingredient of the proof is that we establish a tighter non-asymptotic bound relating the leverage scores for polynomial regression to the Chebyshev density (Theorem C.3).

---

> > ### Comment · Reviewer_Q8k3 · 2023-11-23
> >
> > Dear authors,
> >
> > Thank you for your precise answer. I would raise my score accordingly and continue to support accepting the paper.

---

### Author Response · Authors · 2023-11-18

We would like to thank the reviewers for their supportive response to our work, and a number of thoughtful suggestions for how to improve the presentation of the paper. We have responded to all individual questions below, but please let us know if any other questions arise.

---

### Meta-Review · Area_Chair_zJi2 · 2023-12-11

**Metareview:**

This work presents a novel approach to the problem of active learning for linear regression with an unknown right-hand side. The motivation of the work is to develop a method that balances between the empirical advantages of grid-based deterministic sampling methods that work well in practice because they cover the domain well, and the theoretical approximation guarantees available for techniques based on leverage score sampling.

The main technical contributions of the paper are: 1) a result stating that any sampling method that samples from the rows of the matrix with marginal probabilities given by the leverage scores of the rows, and ensures a certain weakened independence condition will match the sample complexity of levarage score sampling, and 2) the introduction of a novel pivotal sampling technique with these properties, that therefore has the same sample complexity as leverage score sampling for this problem and empirically works better in practice due to increased spatial coverage. Additionally, the authors establish that pivotal sampling has a lower sample complexity than leverage score sampling for polynomial regression.

The reviewers found the paper to be well-written, sound, and to make important contributions; for these reasons, acceptance is recommended.

**Justification For Why Not Higher Score:**

N/A

**Justification For Why Not Lower Score:**

The paper develops a novel active learning approach with the best properties of deterministic and randomized sampling schemes.

---

### Decision · Program_Chairs · 2024-01-16

Accept (oral)